# Molecular and Biological Investigation of Isolated Marine Fungal Metabolites as Anticancer Agents: A Multi-Target Approach

**DOI:** 10.3390/metabo13020162

**Published:** 2023-01-21

**Authors:** Hanin A. Bogari, Sameh S. Elhady, Khaled M. Darwish, Mohamed S. Refaey, Radi A. Mohamed, Reda F. A. Abdelhameed, Ahmad J. Almalki, Mohammed M. Aldurdunji, Manar O. Lashkar, Samah O. Alshehri, Rania T. Malatani, Koji Yamada, Amgad I. M. Khedr

**Affiliations:** 1Department of Pharmacy Practice, Faculty of Pharmacy, King Abdulaziz University, Jeddah 21589, Saudi Arabia; 2Department of Natural Products, Faculty of Pharmacy, King Abdulaziz University, Jeddah 21589, Saudi Arabia; 3Department of Medicinal Chemistry, Faculty of Pharmacy, Suez Canal University, Ismailia 41522, Egypt; 4Department of Pharmacognosy, Faculty of Pharmacy, University of Sadat City, Sadat City 32897, Egypt; 5Department of Aquaculture, Faculty of Aquatic and Fisheries Sciences, Kafrelsheikh University, Kafrelsheikh 33516, Egypt; 6Department of Pharmacognosy, Faculty of Pharmacy, Galala University, New Galala 43713, Egypt; 7Department of Pharmacognosy, Faculty of Pharmacy, Suez Canal University, Ismailia 41522, Egypt; 8Department of Pharmaceutical Chemistry, Faculty of Pharmacy, King Abdulaziz University, Jeddah 21589, Saudi Arabia; 9Department of Clinical Pharmacy, College of Pharmacy, Umm Al-Qura University, P.O. Box 13578, Makkah 21955, Saudi Arabia; 10Garden for Medicinal Plants, Graduate School of Biomedical Sciences, Nagasaki University, Bunkyo-machi 1-14, Nagasaki 852-8521, Japan; 11Department of Pharmacognosy, Faculty of Pharmacy, Port Said University, Port Said 42526, Egypt

**Keywords:** Red Sea fungi, *Penicillium chrysogenum*, indole-based alkaloids, anticancer activity, Cdc-25A, PTP-1B, c-Met kinase, molecular modelling, drug-likeness/pharmacokinetic profiling

## Abstract

Cancer is the leading cause of death globally, with an increasing number of cases being annually reported. Nature-derived metabolites have been widely studied for their potential programmed necrosis, cytotoxicity, and anti-proliferation leading to enrichment for the modern medicine, particularly within the last couple of decades. At a more rapid pace, the concept of multi-target agents has evolved from being an innovative approach into a regular drug development procedure for hampering the multi-fashioned pathophysiology and high-resistance nature of cancer cells. With the advent of the Red Sea *Penicillium chrysogenum* strain S003-isolated indole-based alkaloids, we thoroughly investigated the molecular aspects for three major metabolites: meleagrin (MEL), roquefortine C (ROC), and isoroquefortine C (ISO) against three cancer-associated biological targets Cdc-25A, PTP-1B, and c-Met kinase. The study presented, for the first time, the detailed molecular insights and near-physiological affinity for these marine indole alkaloids against the assign targets through molecular docking-coupled all-atom dynamic simulation analysis. Findings highlighted the superiority of MEL’s binding affinity/stability being quite in concordance with the in vitro anticancer activity profile conducted via sulforhodamine B bioassay on different cancerous cell lines reaching down to low micromolar or even nanomolar potencies. The advent of lengthy structural topologies via the metabolites’ extended tetracyclic cores and aromatic imidazole arm permitted multi-pocket accommodation addressing the selectivity concerns. Additionally, the presence decorating polar functionalities on the core hydrophobic tetracyclic ring contributed compound’s pharmacodynamic preferentiality. Introducing ionizable functionality with more lipophilic characters was highlighted to improve binding affinities which was also in concordance with the conducted drug-likeness/pharmacokinetic profiling for obtaining a balanced pharmacokinetic/dynamic profile. Our study adds to the knowledge regarding drug development and optimization of marine-isolated indole-based alkaloids for future iterative synthesis and pre-clinical investigations as multi-target anticancer agents.

## 1. Introduction

Cancer is a leading cause of death worldwide. In emerging countries, the prevalence of cancer-related death is increasing at an alarming rate [1]. Within global bases, cancer is currently recognized as the second main reason for death with estimated 10 million causalities only in 2020, among which cancers of the lungs, breasts, colon, prostate, rectum, gut, and skin were the most commonly reported [2]. Notably, lung cancer-associated fatalities were responsible for almost 20% of the all cancer-related mortalities in 2020 through global statistical analysis [2]. Within the past two decades, advances within three-dimensional visualization, multimodal imaging, combinatorial therapies, and nano-based medicines have improvised the cancer management drug [3]. Additionally, progress within chemotherapy, cancer-target immunotherapy, epigenomic, and gene therapies has enabled substantial effective control of cancer progression [4]. Despite the healthcare breakthroughs, the continuous uprising of cancer resistance has hampered several cancer therapies, being the principal limiting factor against curing cancer patients as well as contributing within the financial toxicity of cancer care reaching up to USD 207 billion in 2020 [5]. It was not until the 2000s that the multi-target drug approach was adopted for overcoming cancer resistance, reducing chemical space for managing cancer, as well as the cost of developing them [6]. The design of multi-target drugs which are agents that can simultaneously interact with multiple and different biological targets for the treatment of multifactorial pathologies represents a novel interesting approach and a new challenge in medicinal chemistry [7]. In 2007, the first multi-target anticancer drug, sorafenib, gained its US-FDA approval for its significant inhibition of cancer progression and angiogenesis through targeting several protein kinases; CD117/c-Kit, platelet-derived growth factor receptor-beta (PDGFR-β), rapidly accelerated fibrosarcoma (Raf), vascular endothelial growth factor receptor (VEGFR)-2, and VEGFR3 [8]. Developing drugs binding to diverse biological targets has been pursued for the last 3 to 4 decades with the chemical space being the principal limiting factor for developing novel agents [2].

Discovery and development of new amended chemotherapeutics obtained from natural origin has been a modern advancement in cancer remediation owing to their high activities, better tolerability profiles, as well as providing wide chemical space for future manipulation and optimization [9]. Several nature-derived core molecules were reported with multi-target potentialities including: the marine-isolated pyrano-quinolone scaffold as inhibitor of both cyclooxygenase (COX)-2 and EGFR signaling pathway [10]. Multiple blockages of the phosphatidylinositol-3-kinases (PI3K), adenomatous polyposis coli (APC), and retinoblastoma protein (RB) signaling was also reported with taxol and vincristine, the plant-origin anti-mitotic agents [11]. Anticancer potentiality of natural compounds derived from endophytic fungus has also been recognized as ongoing and abundant sources of anticancer agents [12]. Secondary metabolites generated from marine-derived fungi have received a lot of attention in recent years, since many of them are architecturally inimitable and have interesting biological and multi-path pharmacological profile [13]. Among these diverse fungi, *penicillium* is the most prevalent hyphomycete [14]. The pharmacological and biological effects of these fungal secondary metabolites have been documented to include anti-inflammatory, antifungal, antibacterial, anticancer, immunosuppressive, and cholesterol-lowering capabilities [15]. *Penicillium chrysogenum* (*P. chrysogenum*) is a mold that is regularly discovered as a food spoiling agent, and its usage in the synthesis of the antibiotic penicillin has garnered a lot of interest [16].

Reported studies illustrated the in vitro inhibition activity of *P. chrysogenum*-isolated metabolites, particularly its indole-based alkaloids, against a number of cancer-related biological targets [17,18]. These indole-based alkaloids, recognizably meleagrin, were reported with high inhibition ratio (96.6%) against human cell division cycle, Cdc-25A, phosphatase at 100 μM [17]. This Threonine/Tyrosine dual-specific phosphatase enzyme target is an important cell cycle regulator highly expressed throughout G1-to-S and G1-to-M cell stage transitions being responsible for promoting S- and M-phase entry as well as early/late cell cycle progression into mitosis [19]. Tumorigenesis and poor clinical progression are generally promoted under Cdc-25A overexpression within various cancer types including thyroid, ovarian, hepatocellular, colorectal, esophageal, and laryngeal carcinoma as well as non-Hodgkin’s lymphoma [20,21,22]. The important role of Cdc-25A as cell cycle checkpoint as well as deregulator of cycle proteins down raise great interests for discovering small molecules that would hamper oncogenesis [21].

Concerning another interesting cancer-related biotarget, a recent study by Han et al. highlighted the in vitro inhibition profile of *P. chrysogenum* marine fungus-isolated 15 metabolites, on human cytoplasmic PTPase member, namely protein tyrosine phosphatase-1B (PTP-1B) [18]. Within the last decades, PTP-1B arose as one of the newly pursued anticancer therapeutic targets for illustrating significant positive regulation of the erythroblastic oncogene B-2 (ErbB-2, a.k.a. human epidermal growth factor receptor-2 (HER-2)) and cellular sarcoma (Scr)-induced cellular signaling [23,24,25,26]. Major challenges are considered for developing *h*PTP-1B inhibitors since the active site bears high charge density as well as being conserved across the PTPase family that would raise issues of off-target effects [27,28]. Addressing these challenges, strategies focused on developing extended compounds capable of occupying the active site as well as neighboring clefts of PTPase non-conserved residues [27,29]. Others aimed to develop inhibitors of balanced polar/hydrophobic profiles capable of occupying the pocket while achieving good cell permeabilities [30]. Developing of such small molecules has been considered as a successful strategy to improve target’s inhibition activity and maximize on-target affinity [27].

Moving again toward another promising cancer-related biotarget, Mady et al. reported the anti-proliferative activity of *P. chrysogenum*-isolated alkaloid on six human breast cancer cell lines as well as excellent inhibitory activity on human c-Met kinase/hepatocyte growth factor (c-Met/HGF) [31]. The authors provided in vitro experimental evidence for *P. chrysogenum*-isolated indole alkaloids’ activities on c-Met as a promising anti-breast cancer agent. The investigated biotarget has been long recognized as the dysregulated signaling pathway involved within breast carcinoma pathogenesis activating the downstream signaling pathways including PI3K/Akt, Ras/MAPK, and c-Src signaling [32]. Dysfunctional c-Met/HGF signaling has been evident virtually within all solid tumors being involved in multiple tumor oncogenic processes including cellular survival, mitogenesis, invasive proliferation, angiogenesis, and metastasis [33,34,35]. Correlation to poor clinical progression and metastatic prognosis was demonstrated with c-Met/HGF overexpression in several human tumors in prostate, kidney, liver, ovarian, lung, and gut [36]. Therefore, c-Met inhibition via receptor blockers were found beneficial for hampering tumor cell line motility, tumorgenicity, and invasiveness [37]. Reported efficiencies of marketed c-Met inhibitors (cabozantinib and crizotinib) as well as those under clinical trials further validate c-Met as a cancer-related therapeutic biotarget encouraging further study of new promising inhibitors [36,38].

Based on the presented evidence, indole-based alkaloids showed potentiality as multi-target agent hampering cancer invasion and cellular viability. Despite the promising therapeutic avenue, designing a multi-target drug is considered challenging regarding the required activity ratios and adequate selectivity at the discovery stage [4,6]. In such a field, computational modelling offers a facilitated real-time reference guide for compound’s target selectivity and off-target profiling, as well as rationalized optimization either through ligand- or structural-based tactics. For the benefit of advanced cheminformatics and bioinformatics, as well as sophisticated algorithms and software generations, computational approaches greatly contribute to drug discovery and development at more rapid paces and lower expenders than traditional methods [39]. Here, we firstly present a comprehensive molecular insight for meleagrin, as well as its close-structural analogues roquefortine C and isoroquefortine C all isolated from *P. chrysogenum* strain S003, regarding their affinity/binding with the three cancer-related biological targets. None of the above reported *P. chrysogenum*-oriented studies have provided comprehensive investigation of the isolated indole metabolite with respect to their molecular bases. The detailed molecular modelling insights and near-physiological simulation for metabolites/target binding affinity/interactions that would be beneficial for future lead development and optimization are needed. Thus, through our adopted sophisticated molecular modelling approach of molecular docking-coupled explicit dynamic simulation and drug likeness/pharmacokinetic predictions, valuable insights were represented which would be best implemented in future candidate structural optimization and clinical development. Additionally, findings from the computational studies showed relevant coherence with the conducted in vitro biological assay on different cancerous cell lines in terms of the compounds’ comparative activities. Altogether, our presented manuscript is highly valuable to the authors of previous works and to the continuous efforts for providing molecular insights and future guidance towards the optimization and further clinical investigation of natural-driven lead compounds.

## 2. Experimental Design

### 2.1. Fungal Materials

*P. chrysogenum* strain S003 was cultured from the Red Sea sediment and the fungal strain was identified as previously described [40].

### 2.2. Fermentation, Extraction, Isolation, and Purification of Compounds **1**–**3**

Fungus *P. chrysogenum* strain S003 was cultured on Czapek–Dox yeast extract liquid medium at room temperature in 2 L Erlenmeyer flasks under static conditions for 30 days. The fermented whole broth (15 L) and the mycelia were extracted and partitioned between different solvents and dried to afford the crude extract. The crude extract was chromatographed over different stationary phases including silica gel, sephadex LH-20 and Octadecylsilane (ODS; Nacalai, Inc., Kyoto, Japan, (0.5 × 10 cm, id)) silica gel column to afford meleagrin (MEL), roquefortine C (ROC), and isoroquefortine C (ISO). Detailed information on the extraction, isolation, and purification as well as description list of the MS and NMR instruments used in structural elucidation are within Appendix A.

### 2.3. In Vitro Activity

Activity profile of the purified compounds on hepatocellular carcinoma HepG2, prostate cancer DU-145, cervical cancer HeLa, and lung cancer A-549 was determined using sulforhodamine B (SRB) assay as per reported method [41]. The adopted technique is a cell density determination assay widely used for chemosensitivity testing and cell proliferation analysis [42].

### 2.4. Molecular Docking

Ligands were constructed via the isomeric SMILES strings Using the AutoDock Vina 1.2.0 software suit (Scripps Research, La Jolla, CA, USA) [43,44]. Ligands were optimized through merging the non-polar hydrogens and Gasteiger charges, before the ligands were energy minimized for optimizing bond distances and angles. Ligand structures were then transformed into pdbqt.file formats. Biological targets; c-Met kinase (PDB: 5ya5), Cdc25A (PDB: 1c25), and PTP-1B (PDB: 4i8n) were structurally prepared through water/solvent/ion removal, Gasteiger charges computation, polar hydrogens addition, and non-polar hydrogens merged using AutoDock Vina software. Binding sites were defined by endorsement of co-crystallized ligands as well as being refined to include the crucial residues reported in current literature as being thoroughly described within the manuscript. Docking workflow was performed under Vina Forcefield, while as the ligand conformational search was performed through Lamarckian Genetic Algorithm, and Genetic algorithm was assigned for docked binding pose predictions [43,44]. The center of biological target within its crystalline structure was set as the docking box center. Global search exhaustiveness was set at eight and maximum energy differences between binding modes was defined at three Kcal/mol [45]. High docking scores, RMSD values below 2.0 Å cut-off, and/or significant interactions with reported crucial pocket residues were considered for selecting the best ligand’s docking pose. PyMol2.0.6 (Schrödinger, New York, NY, USA) was used for visualization and binding interaction analysis [46].

### 2.5. Molecular Dynamics Ensembles

GROMACS-2019 software package under CHARMM36m force field for protein and CHARMM-General Force Field program (Param-Chem project; https://cgenff.umaryland.edu/: accessed on 20 July 2022) for ligands, was used to conduct the explicit molecular dynamics simulations [47,48]. Ligand–protein model was solvated within TIP3P cubic box under periodic boundary conditions with 10 Å marginal distances [49]. Protein residues of were assigned at their standard ionization states (pH 7.4), while the entire system net charge was neutralized via potassium and chloride ions [50]. Constructed systems were minimized through 5 ps under the steepest descent algorithm [51], and they were then equilibrated for 100 ps under NVT ensemble (303.15 K) followed by 100 ps NPT ensemble (1 atm. Pressure and 303.15 K) [52]. The production stage involved 100 ns MD simulation runs under NPT ensemble while using the Particle Mesh Ewald algorithm for computing the long-range electrostatic interactions [53]. All covalent bond lengths were modeled under LINCS with 2 fs integration time step size [54]. Both Coulomb’s and van der Waals non-bonded interactions were truncated at 10 Å using the Verlet cut-off scheme [55]. The binding-free energy between the ligand and protein as well as the residue-wise contributions within the binding-free energy calculations were estimated via MM/PBSA on representative frames for the whole MD simulation runs (100 ns) [56].

### 2.6. Drug-Likeness and Pharmacokinetic Profiling

Both drug-likeness and clinical fitness of the isolated metabolites were evaluated via Qik_Prop^®^ V3.5 (Schrödinger, New York, NY, USA) and TEST^®^_Toxicity Estimation Software Tool^®^ version-4.2.1 (Environmental Protection Agency, Washington, DC, USA). Adherence of the investigated compounds towards the Lipinski’s rule of five (R_O5) was adopted to evaluate the compound’s drug-likeness profile. The rule is defined by estimating the molecular and structural properties/descriptors including: number of hydrogen bond donors (HBDs) ≤ 5, hydrogen bond acceptors (HBAs) ≤ 10; rotatable/torsion bond (θ) number < 5, molecular weight (MW) < 500 daltons and Q_logP_o/w_ < 5 loop. Lipinski’s R_O5 is considered as the gold standard for drug-likeness and ADME assessment, yet not a strict criterion for natural products [57,58]. However, a natural compound exhibiting minimal or preferably lacking R_O5 violations is more likely to be a promising clinical candidate with oral bioavailability with fewer attritions throughout prospective clinical trials [59,60].

Regarding the pharmacokinetic profiling, the adopted Qik_Prop^®^ module permits an accurate prediction of several significant physical descriptors and main pharmaceutical-relevant properties as being related to the ADME_TOX properties [61]. The pharmacokinetic_ADME-related physio/chemical properties includes: partition coefficient in octanol-water system (Q_PlogP_o/w_), solubility in aqueous media (Q_PlogS), apparent permeability through Caco2-cells (Q_PPCaco) modelling the passive transportation across the blood/gut barrier, partition coefficient between blood and brain (Q_PlogBB) modelling the blood brain barrier penetration and CNS access, apparent permeability across kidney cells of Madin_Darby dog (Q_PPMDCK) modelling the blood/brain barrier, binding to human serum albumins (Q_PlogK_HSA_), and finally human’s percent oral absorption (%HOA) [62,63,64,65,66]. Evaluating the compound’s toxicological profile was proceeded through predicting the half-maximal inhibitory concentration (IC_50_) for blocking HERG_K_v_11.1-channels (Q_PlogHERG), Oral lethal dose 50 (LD_50_) on rats, as well as AMES_Mutagenicity test via TEST^®^-consensus approach [67]. Both LD_50_ and Q_PlogHERG represent the compound’s concentration (mg/Kg) needed for respective 50% rat death after a per oral administration or positive induction of colony growth with any *Salmonella typhimurium* strain.

## 3. Results and Discussion

### 3.1. Isolation, Purification, and Structural Identification of Compounds ***1**–**3***

The chromatographic separation of the crude extracts obtained from culture broth and mycelia of fungus *P. chrysogenum* strain S003 using silica gel and Sephadex LH-20 column chromatography resulted in the isolation of three pure metabolites **1**–**3**, (Figure 1). By comparing the detailed NMR spectral data with those in the literature the compounds were determined to be MEL (**1**) [68,69], ROC (**2**) [68,69], and ISO (**3**) [69].

**Meleagrin (1):** It was obtained as light yellow crystals, C_23_H_23_N_5_O_4_ and positive FAB-MS at *m*/*z* 434 [M+H]^+^; ^1^H-NMR (400 MHz, CDCl_3_) *δ*_H_: 7.59 (1H, d, *J* = 7.8 Hz, H-4), 7.08 (1H, dt, *J* = 7.5 Hz, H-5), 7.26 (1H, dt, *J* = 7.8 and 0.95 Hz, H-6), 6.98 (1H, d, *J* = 7.8 H z, H-7), 5.37 (1H, s, H-8), 8.29 (1H, s, H-15), 7.34 (1H, s, H-18), 7.78 (1H, s, H-20), 6.12 (1H, brs, H-22), 5.03 (1H, d, *J* = 17.1, Ha-23), 5.08 (1H, d, *J* = 9.6, Hb-23), 1.23 (3H, s, H-24), 1.27 (3H, s, H-25), and 3.74 (3H, s, 1-OCH_3_). ^13^C-NMR (125 MHz, CDCl_3_:CD_3_OD (1:1)) *δ*_C_: 101.6 (C-2), 52.5 (C-3), 126.1 (C-3a), 124.9 (C-4), 122.7 (C-5), 128.5 (C-6), 111.9 (C-7), 146.4 (C-7a), 109.5 (C-8), 141.8 (C-9), 159.2 (C-10), 123.6 (C-12), 165.6 (C-13), 108.5 (C-15), 125.4 (C-16), 134.3 (C-18), 136.8 (C-20), 42.6 (C-21), 141.8 (C-22), 111.9 (C-23), 29.7 (C-24 and 25), and 65.4 (1-OCH_3_) (Appendix A).

**Roquefortine C (2):** It was obtained as yellowish white solid, C_22_H_23_N_5_O_2_, and positive FAB-MS at *m*/*z* 390 [M+H]^+^; ^1^H-NMR (400 MHz, CDCl_3_) *δ*_H_ 9.25 (1H, s, H2), 5.70 (1H, s, H5a), 5.03 (1H, s, H6), 6.67 (1H, d, *J* = 7.8 Hz, H7), 6.78 (1H, t, *J* = 7.5 Hz, H8), 7.17 (1H, t, *J* = 7.7 Hz, H8), 7.25 (1H, d, *J* = 7.6 Hz, H10), 2.53 (1H, t, *J* = 11.9 Hz, H11), 4.13 (1H, dd, *J* = 6.0, 11.4 Hz, H11a), 2.67 (1H, dd, *J* = 6.0, 12.4 Hz, H11), 6.35 (1H, s, H12), 7.33 (1H, s, H15), 7.76 (1H, s, H17), 6.05 (1H, dd, *J* = 10.8, 17.4 Hz, H19), 5.19 (2H, m, H20), 1.21 (3H, s, H-21), and 1.09 (3H, s, H-22). ^13^C-NMR (100 MHz, CDCl_3_) *δ*_C_ 166.9 (C-1), 121.4 (C-3), 159.4 (C-4), 78.4 (C-5a), 149.8 (C-6a), 111.4 (C-7), 129.1 (C-8), 119.1 (C-9), 125.1 (C-10), 128.5 (C-10a), 61.5 (C-10b), 38.7 (C-11), 58.8 (C-11a), 109.1 (C-12), 125.1 (C-13), 136.9 (C-15), 135.3 (C-17), 40.9 (C-18), 143.3 (C-19), 114.8 (C-20), 22.4 (C-21), and 22.8 (C-22) (Appendix A).

**Isoroquefortine C (3):** It is obtained as yellowish white solid, C_22_H_23_N_5_O_2_, and positive FAB-MS at *m*/*z* 390 [M+H]^+^. ^1^H-NMR (400 MHz, CDCl_3_) *δ*_H_ 11.58 (1H, s, NH), 9.38 (1H, s, NH), 4.94 (1H, s, NH), 5.66 (1H, s, H5a), 6.58 (1H, d, *J* = 7.8 Hz, H7), 6.75 (1H, t, *J* = 7.4 Hz, H8), 7.09 (1H, td, *J* = 7.7, 1.4 Hz, H9), 7.19 (1H, m, H10), 2.49 (1H, t, *J* = 11.9 Hz, H11), 4.12 (1H, dd, *J* = 5.8, 11.4 Hz, H11a), 2.61 (1H, dd, *J* = 5.8, 12.2 Hz, H11), 6.72 (1H, s, H12), 7.69 (1H, s, H15), 7.16 (1H, s, H17), 6.00 (1H, dd, *J* = 10.9, 17.3 Hz, H19), 5.12 (2H, m, H20), 1.15 (3H, s, H-21), and 1.04 (3H, s, H-22). ^13^C-NMR (100 MHz, CDCl_3_) *δ*_C_ 165.7 (C-1), 137.9 (C-3), 158.6 (C-4), 78.1 (C-5a), 150.5 (C-6a), 109.1 (C-7), 129.1 (C-8), 118.9 (C-9), 125.4 (C-10), 129.1 (C-10a), 61.7 (C-10b), 37.2 (C-11), 59.2 (C-11a), 104.9 (C-12), 127. (C-13), 135.2 (C-15), 117.1 (C-17), 41.0 (C-18), 143.8 (C-19), 114.7 (C-20), 22.5 (C-21), and 23.0 (C-22) (Appendix A).

### 3.2. In Vitro Activity of Isolated Compounds ***1**–**3***

The isolated indole-based compounds (MEL, ROC, and ISO) were evaluated for their inhibition activity in terms of cancerous cell number and viability using the cell metabolic activity assay; sulforhodamine B (SRB), compared with doxorubicin (DOX) positive control (Table 1). MEL displayed the most promising activity profile with IC_50_ values at low and even sub-micromolar micromolar concentrations. Notably, MEL depicted preferential activity profiles against both prostate DU-145 and breast HepG2 cancer cell lines with respective IC_50_ values at two- and three-digit nanomolar potencies. ROC depicted moderate (two-digit micromolar) activities against lung A-549 and cervical Hela cancerous cell lines, while as the other two cancer cells were assigned with lower micromolar activity profiles. Finally, the isomeric metabolite, ISO, showed a modest activity profile since the treated cell line was illustrated with higher micromolar activities. Activity for ISO against the cervical cancer HeLa cells was the fairest (>50 μM). Notably, HeLa cell line was considered of potential resistance towards both ROC and ISO with respective activity profiles above 40 μM and 50 μM activity concentrations. It is worth noting that only MEL managed to achieve lower nanomolar inhibition activities at both DU-145 and HepG2 compared with those of the positive control DOX.

### 3.3. Multi-Target Docking Analysis on Cancer-Associated Molecular Biotargets

#### 3.3.1. Docking Simulation on Cdc-25A

A validated molecular docking simulation was conducted for explaining the molecular bases regarding the in vitro activity of MEL and related analogues on tested cancerous cells. Structural features and topology of Cdc-25A catalytic domain involved a small-sized α/β structure composed of a solvent-exposed binding active site and two structural motifs, CH2A and CH2B (Figure 2A). This phosphatase enzyme shares the Cys-(x)_5_-Arg motif being common with several tyrosine kinase proteins [70,71]. The catalytic Cys430 serves as a nucleophile projecting into the center of the loop harboring the Cys-(x)_5_-Arg P-loop motif and connecting between an α-helix and a β-sheet. The last Arg436 motif is responsible for anchoring and stabilizing the phosphate within the transition state. The peptide amides of the five middle residues mediate hydrogen bond-directed stability for the substrate phosphate group [72]. Neighboring Ser433 is suggested fundamental for stabilizing the catalytic Cys430 thiol group in order to assist the second catalytic step. Additionally, His429 sidechain offers relevant stability for the Cys430 mainchain through hydrogen bonding and polar contact [19,73]. Recognizing and directing the tyrosine-protein substrates into the catalytic pocket is achieved through the hydrophobic-directed advent of the Phe229 sidechain [74,75]. Unlike other phosphatase enzymes, Glu rather than Asp showed relevant proximity towards the active site for catalysis assistance through serving as general acid (proton donor to the leaving hydroxyl group) [76]. In all other phosphatases, Asp general acids are on flexible loop far from the active site [77]. Additionally, Asp383 in Cdc-25A is directed off-plane from the active site and considered conserved for structural rather than mechanistic purposes [78]. Thus, targeting Glu431 in Cdc-25A for ligand binding would provide target selectivity and minimized off-target toxicity as ligands would dodge other phosphatase target inhibition.

Docking of MEL and related analogues showed preferential accommodation for some ligands over the others within the Cdc-25A binding site within the inner space between the Cys-(x)_5_-Arg, CH2A, and CH2B motifs (Figure 2B). Compounds laid their tetracylic scaffold near the Cys-(x)_5_-Arg motif while directing their terminal imidazole rings towards the solvent side. Meleagrin showed the closest proximity towards the catalytic Cys-(x)_5_-Arg motif via its aromatic dihydroindole fused ring. On the other hand, roquefortine C (ROC) settled its aromatic ring with relevant proximity towards the CH2A motif. Despite the latter differential indole ring orientations, both imidazole rings of MEL and ROC were at comparable orientations being superimposed near the β-sheet comprising the catalytic active site. Unlike ROC, the isomeric analogue isoroquefortine C (ISO) depicted different orientation for its terminal imidazole-based arm being more directed towards the solvent side near the terminal *C*-terminal β-loop of CH2A motif. The ROC/ISO differential orientations can be the reason for the varied configurations of the methylbutylidene moieties imposing differential steric clashes with the lining residues of the Cys-(x)_5_-Arg motif. Aside from steric preferentiality, differential ligand–protein binding patterns were also depicted significant for explaining the specific orientation/conformation for each docked ligand within the Cdc-25A active binding site.

Interestingly, docking of MEL within the binding site is guided through extensive polar interactions with hydrophilic residues of the Cys-(x)_5_-Arg and active site β-sheet. Stability of the MEL-protein complex is mediated through double hydrogen bonding with each of Glu431 (bond distance/angle; 1.8 Å/163.3°; 2.5 Å/127.1°), Arg436 (2.2 Å/140.1°; 3.2 Å/129.4°), and Arg439 (2.5 Å/129.6°; 2.8 Å/121.7°) through the three oxygen functionalities within the MEL structure being served as H-bond donors and/or acceptors (Figure 2C). Anchoring of the ligand’s aromatic dihydroindole ring is driven via favored hydrophobic contacts with Phe432 (5.3 Å). Despite being charged under physiological conditions, both Glu431 and Glu435 depicted reasonable hydrophobic van der Waal interactions via their sidechain Cβ and Cδ with the MEL’s α-oriented methylbutylidene moiety (~ 4.1 Å) and aromatic dihydroindole ring (~3.6 Å), respectively. These hydrophobic interactions further favored the MEL deep anchoring at the Cdc-25A binding site. Finally, docking the MEL’s terminal imidazole ring near the active site sheet was favored through π–H non-classical hydrogen bond interaction with the Arg439 sidechain hydrogen (4.3 Å). All represented MEL’s docking features would be successfully translated into its respective high docking score (−8.98 Kcal/mol).

Stability of ROC within the active site was depicted through a combined polar and hydrophobic binding interaction. Hydrogen bond pairing with the polar residue of Cys-(x)_5_-Arg motif were illustrated via the ligand’s piperazindione moiety (carbonyl and protonated N atom) (Figure 2D). Serving as hydrogen bond acceptor, the ring’s carbonyl group furnished single hydrogen bonding with Ser434 mainchain NH (3.2 Å/145.3°) and catalytic Cys430 sidechain (3.1 Å/125.2°). Stability of ROC at the catalytic Cys-(x)_5_-Arg motif was further fortified through hydrogen bonding with Ser434 mainchain carbonyl moiety (2.5 Å/124.7°). Showing similar MEL orientation for its terminal imidazole, ROC depicted favorable distance and contact with the active site Arg439 to furnish relevant π–H bonding interaction (3.4 Å). On the other hand, the differential orientation of ROC’s aromatic dihydroindole fused ring illustrated proper orientation and close distance towards Tyr386 of CH2A motif furnishing significant π-stacking hydrophobic interaction (4.6 Å). Despite lacking relevant polar interaction with Arg436 sidechain, the latter residue depicted closeness towards the polar functionalities decorating the piperazine scaffold (~4.0 Å) which would suggest reasonable stability through water-bridge bonds. Finally, the ROC’s β-oriented methylbutylidene moiety showed close proximity towards Phe432 (4.1 Å) and Glu431 β/δ-carbons (~4.2 Å) mediating relevant non-polar contacts. It worth noting that ROC depicted a lower range of polar ligand–target interactions which can be due to its obtained moderate docking binding energy (−7.95 Kcal/mol).

Moving towards the ISO binding mode, it was clear that isomeric analogue imposed significant impact in orienting the ligand at the Cdc-25A binding site. In compared orientation to ROC, the α-oriented methylbutylidene moiety at ISO would impose significant clashes with Glu431 and vicinal residue. In this regard, a better less steric orientation was adopted for ISO directing its methylbutylidene arm towards a more vacant space at the binding site near to Arg436 side of the Cys-(x)_5_-Arg motif (Figure 2E). The latter methylbutylidene moiety conformation/orientation caused ISO to be settled at closer proximity towards the flexible loop of the CH2A motif. The ligand’s piperazindione ring depicted relevant hydrogen bond pairing with polar functionalities of Tyr386 and His402 sidechains (2.6 Å/129.9° and 2.2 Å/127.5°, respectively). Contrarily to both MEL and ROC, the imidazole terminal ring in ISO lacked relevant closeness and interaction with the active site Arg439 sidechain. Additionally, the ISO aromatic scaffold was distant from neighboring aromatic pocket’s residues which lacked relevant π-mediated ligand–target interactions. Hydrophobic van der Waal contacts between the α-oriented methylbutylidene moiety and Glu431 β/δ-carbons (~4.0 Å) as well as Cys430 sidechain (~3.8 Å) can favor ligand–target complex stability to some extent. All those findings were correlated to modest docking energy for ISO (−6.49 Kcal/mol) compared with the other two analogues.

Validation of the obtained docking poses were ensured through redocking procedures furnishing low root-mean square deviations; RMSDs of 1.30 Å, 1.46 Å, and 0.98 Å for the superimposed docked/redocked poses of MEL, ROC, and ISO, respectively. Depicting rRMSD numbers below the 2.0 Å threshold would highlight the validity of the adopted docking protocol as well as ensure the biological significance of the obtained docking binding modes and in turn their respective scored energies [47,79,80,81,82]. Additionally, the obtained ligand binding poses were further confirmed valid through the ability of docked ligands to replicate the reported residue-wise binding patterns of small molecular ligands reported within the current literature. Through structure-based virtual screening, Park et al. discovered two triazole-based small molecules with promising affinity and Cdc-25A inhibition activity (IC50 = 0.82–4.07 μM) [83]. Both triazole-based compounds mediated hydrogen bonding with the P-loop Glu431 sidechain via their catecholcarbonyl moieties, while maintaining vicinity with their triazole ring at the Arg436 amino acid mimicking the substrate phosphate group. Hydrophobic interactions with Tyr386, Trp507, as well as π-cation with Arg401 were found important to partially compensate the electrostatic penalty of ligand–solvent exposure. Importance of Glu431-directed binding was further highlighted by Park et al. through E431A point mutation leading to reduced stability and binding free energies of tested triazoles. LigBuilder-probed de novo drug design identified cyano-substituted thioisocytosine-based hits able to depict deep docking at the Cdc-25A active site and promising inhibitory activity (IC50 = 2.30–6.70 μM) [84]. Polar interactions with Ser434 mainchain and Arg436 sidechain besides π-mediated hydrophobic contacts with Arg439 and Tyr386 highlighted the significance of these residues for thioisocytosine-based hits stability. The same residue-based binding patterns were also signified within a pharmacophore-guided molecular docking drug discovery program by Ferrari et al. identifying naphthoquinone-derived hits with Cdc-25A/B inhibition activity (IC50 = 0.89–1.42 μM), tumor regression, and cell cycle arrest capabilities [85].

#### 3.3.2. Docking Simulation on PTP-1B

Exploring the molecular bases of MEL and related analogues binding at *h*PTP-1B, active sites and vicinal pockets were pursued through a valid molecular modelling simulation. Identified as the P-loop, the *h*PTP-1B residue range His214-Arg221 comprises the substrate’s central binding site which is surrounded by four secondary structured loops including the neighboring phosphotyrosine-recognizing (PTR) loop (Asp40-Tyr46), R-loop (Leu110-Cys121), WPD-loop (Thr177-Pro185), and Q-loop (Glu262-Phe269) (Figure 3A). The conserved Cys215 in PTP-1B is located at the P-loop bottom region together with horseshoe fashion orientation of the backbone amides of neighboring residues [86]. Both the amides and conserved Arg221 hydrogen bond the substrate phosphate group mediating stability for the transition state [87]. Binding induces PTP-1B protein conformational changes being culminated via closure orientation of WPD-loop that harbors the conserved Asp181 acting as general acid and base at the first and second catalysis steps, respectively [88]. Substrate specificity is further contributed by two surface residues, Tyr46 and Arg47, where targeting them for small molecule binding can hamper substrate affinity [87]. A second non-PTPase conserved aryl phosphate (Tyr-P) binding site identified by Puius et al. is vicinal to the P-loop and comprises the residues Tyr20, Arg24, His25, Ala27, Phe52, Arg254, Met258, and Gly259 [29]. Doubly targeting both P-loop and Tyr-P sites would provide the advent of improved potencies down to nM ranges as well as PTP1B selectivity being non-conserved across PTPase family members [27,89].

Docking MEL and close analogues at the PTP-1B binding site showed differential binding modes in relation to each other as well as to the co-crystallized ligand, IN1834-146C (INC). Docked ligands showed preferential accommodation at the catalytic site (P-loop) via their terminal imidazole rings where they depicted comparable orientation to the carbamoyl group of the co-crystallized ligand (Figure 3B). Nevertheless, the highest superimposition of the ligand’s imidazole with the INC’s carbamoyl group was assigned for MEL compared with its related analogues. On the other hand, the isomeric analogue ISO depicted a retraction of its imidazole-based arm from the deep anchoring within the catalytic site P-loop which can be related to the differential configuration of its methylbutylidene moiety. The three docked ligands directed their methylbutylidene groups towards the PTR-loop where those of MEL and ROC depicted comparable orientations being different from that of ISO ligand. On the other side, the fused ring scaffolds of the docked ligands were directed towards the secondary aryl-phosphate site showing MEL with the closest proximity via its terminal aromatic dihydroindole fused ring. Owing to its more extended structure, the co-crystallized ligand exhibited deeper anchoring within the secondary aryl-phosphate site via its benzoxazole moiety. Interestingly, the co-crystallized ligand INC directed its 4-fluoro phenyl group towards the solvent side which impacted the ligand’s crystallographic atomic displacement depicting high B-factor values [90]. Contrarily, none of the docked ligands showed significant solvent exposure conferring limited flexibility in relation to thermal motion [91].

Docking simulation for MEL furnished relevant docking binding energy (−7.98 Kcal/mol; rRMSD 1.37 Å) correlated with favored ligand anchoring. Binding mode of MEL at PTP-1B active sites illustrated several favored hydrogen bonds with the polar lining residues reported with important role within the enzyme catalysis (Figure 3C). The ligand’s imidazole ring mediated polar contacts with the catalytic Cys215 (3.1 Å/122.7°) and vicinal conserved Arg221 (1.9 Å/148.3°) conferring potential MEL-mediated interruption of the PTP-1B’s catalytic machinery. Adopting a closed crystallographic conformation, the WPD-loop polar residue illustrated close proximity towards the MEL’s imidazole ring. The latter favored conformation mediated relevant hydrogen bond pairing between the imidazole’s NH and sidechain of the enzyme’s general acid/base, Asp181 amino acid (1.4 Å/150.2°). Further MEL-pocket stability was mediated through polar contacts between the core ring polar functionalities from one side and the sidechains of Tyr47 (3.3 Å/124.8°) and Gln262 (2.1 Å/157.9°) of the PTR- and Q-loops, respectively. Hydrophobic van der Waal contacts with non-polar residues including Tyr46, Val49, Ala217, Ile219, as well as π–H interaction with Phe182 (2.5 Å) contributed for the anchoring of MEL’s aliphatic/ring scaffolds.

Moving towards ROC docking results, three hydrogen bond interactions were depicted by the ligand’s imidazole head towards the polar residues of the PTP-1B catalytic site (Figure 3D). Serving as both hydrogen bond donor and acceptor, the imidazole nitrogen mediated the hydrogen bond pairing with the P-loop catalytic Cys215 (3.2 Å/119.4°), Arg221 (2.1 Å/126.3°), as well as R-loop Glu115 (1.3 Å/161.2°). Stability of the ligand’s core ring at the PTP-1B binding site was illustrated through polar contact for the carbonyl group of the ROC’s piperazindione scaffold with the hydroxy group of Tyr47 sidechain (3.1 Å/128.6°). In addition to polar interactions, the substituted methylbutylidene group and terminal aromatic dihydroindole fused ring mediated ligand–target complex stability through π-driven hydrophobic interactions with Tyr47 (3.8 Å) and Phe182 (2.8 Å) aromatic sidechains, respectively. Additional van der Waal contacts with Val49, Ala217, and Ile219 added to the ROC binding being all together translated into the relevant docking binding score of −7.74 Kcal/mol (rRMSD 1.55 Å), just comparable to that of MEL ligand.

Regarding the isomeric analogue, ISO, the ligand–target binding mode was greatly defined by the differential configuration of the methylbutylidene group in relation to that ROC. Being at a lower configurational plane, the sterically hindered methylbutylidene group was directed far from the RTP-loop residues (Figure 3E). Such orientation caused both the ligand’s terminal ring to be retracted more far from the Tyr-P secondary site as compared with those of MEL or even ROC. Additionally, the ISO’s imidazole ring was retracted far from the catalytic and important residues of the P-loop site which caused this polar head to lack any contacts with the catalytic Cys215 or vicinal Arg221 sidechains. On the other hand, the ligand’s imidazole scaffold was more directed towards the polar residues of R-loop region furnishing relevant hydrogen bonding with the sidechains of Glu115 (1.6 Å/117.7°) and Lys120 (2.5 Å/122.5°). Interestingly, the ligand’s core ring was further stabilized via hydrogen bond pairing for the piperazine carbonyl group with Ala217 mainchain NH (2.6 Å/140.3°). Despite hydrophobic contacts with Val49 and Ile219, the improper closeness/orientation of the methylbutylidene group and aromatic ring caused ISO to lack relevant π-mediated contacts with Tyr47 or even Phe182 residues. The ISO binding mode revealed the modest binding energy with a docking score of −5.98 Kcal/mol (rRMSD 1.67 Å).

Furnishing low RMSDs for the superimposed native co-crystallized ligand and its redocked pose (1.56 Å; Figure 3F) through the same adopted docking protocol further ensured the validity of the obtained ligand–target binding modes as well as the docking workflow [47,79,80,81,82]. The carboxamide moiety responsible for polar fixation of the ligand at the P-loop showed good superimposition for the docked and co-crystalized ligand. The rest of ligand’s terminal aromatic scaffolds were in right orientations at the Tyr-P subsite replicating the reported hydrophobic interactions of the co-crystallized ligand, despite the little twisted orientation of the *p*-fluorophenyl group. Validation of the docking protocol was further confirmed through depicting residue-wise ligand–target binding interactions consistent with several reported studies introducing small compounds with potential or even actual PTP-1B inhibition activity [92,93,94,95]. The synthesized 4-thiazolinone derivatives introduced by Liu et al. furnished significant PTP-1B inhibition activity with potencies down to one-digit micromolar concentrations (IC50 = 0.92–9.64 μM) with remarkable selectivity profiles over several PTPase targets [92]. Docking of the most active compound at PTP-1B catalytic site revealed residue-wise complex stability via hydrogen bonding with Asp181, Cys215, and Gln262, as well as π-driven contacts with Tyr46 and Lys120. Similar findings were also illustrated for 4-thiazolidine derivatives where through molecular docking investigation polar contacts with Lys120, Arg221, and Gln266 were suggested relevant for complex stability [93]. Subsequent 30 ns molecular dynamic simulation for each top-scored compound highlighted the significant hydrogen bond frequency for the Arg221 residue (>80%) with the simulated ligand. Pai and his research group reported computational investigation of 1941 phytomolecules as potential PTP-1B inhibitors using structurally based molecular docking followed by water mapping and molecular dynamics simulations for the promising hits [94]. Top-scored hits showed significant overlapping with several stable hydration sites based on water-mapping analysis as well as dominant stable hydrogen bonding with Tyr46, Cys215, and/or Arg221 across the 25 ns molecular dynamics simulation runs. Another study explored the binding affinities of nine *Anoectochilus chapaensis*-isolated metabolites with potent inhibitory profiled (IC50 = 1.20–6.21 μM) through docking analysis at PTP-1B catalytic site [95]. Docking analysis revealed hydrogen bonding with Arg221 as the almost consistent ligand–target interaction across the investigated ligand-associated target complexes.

#### 3.3.3. Docking Simulation on c-Met Kinase

For gaining more insights regarding the molecular basis of MEL/c-Met binding affinity, a valid molecular docking simulation was conducted for the drug as well as related analogues to explore the structural-related binding differences. Docking simulation was conducted at the ATP-binding site of the c-Met target (Figure 4A) for mimicking the ATP interactions with the hinge region residues (Pro1158, Tyr1159, and Met1160) being characteristic for all c-Met inhibitors [96]. Binding interactions with activation loop (A-loop) residues, including the kinase conserved DFG motif (Asp1222, Phe1223, and Gly1224) as well as c-Met-specific Tyr1230 and Lys1245 were depicted crucial for kinase receptor antagonism and good selectivity profile, respectively [36,37]. The latter activation loop-oriented binding profile is associated with type-I c-Met inhibitors, such as the FDA-approved crizotinib, where it exhibits relatively good cellular selectivity profile, yet suffer from acquired resistances within the clinic [97,98,99]. On the other hand, inhibitors of type-1.5 and -II binding fashion (e.g., cabozantinib) exhibited deep anchoring into the hydrophobic back pocket beyond the gatekeeper residue (Leu1157) [36,37]. These deep anchoring compounds often possess high potency, yet with poor selectivity profiles being lower than those of type-I inhibitors which would limit the earlier clinical utility [100,101]. Moreover, type-1.5/-II inhibitors are usually highly lipophilic with large molecular weights (often > 500 Da) to furnish relevant interactions at the hydrophobic back pocket for compensating the significant conformational movement of A-loop needed for this deep pocket to be opened [102]. The latter would impose oral bioavailability challenges throughout drug development and optimization stages. Thus, the ability of novel compounds to accommodate the ATP-binding site while depicting relevant contacts with activation loop would impose greater utility as selective c-Met inhibitor with promising pharmacodynamics and non-presidential resistance profiles.

An overlay of the docked indole-based alkaloids over the co-crystallized ligand, 6TD, illustrated relevant accommodation within the c-Met ATP-binding site (Figure 4B). Ligands settled their aromatic imidazole arm at the hinge region while extending their core cyclic scaffold across the ATP-binding site reaching to the A-loop side. The crystalized ligand depicted the typical U-shaped conformation of the type-I c-Met inhibitors [102]. Nevertheless, the docked alkaloids adopted a more curved conformation/orientation at the enzyme pocket owing to the inherited rigidity of their core tetracyclic scaffold. Notably, the imidazolidinone ring in MEL as well as its respective congruent scaffold within ROC and ISO (i.e., piperazindione ring) were well-settled at ATP-adenine binding sub-pocket of the enzyme. Regarding the ligand’s terminal, both benzene rings of Mel and ISO were anchored at the space occupied by the triazolo [3,4-b]thiadiazole scaffold of the co-crystalized ligand. Only MEL’s aromatic scaffold was oriented at relevant co-planar orientation in relation to 6TD heterocyclic fused ring. On the other hand, perpendicular rather than parallel orientation was depicted for the IOS’s benzene ring in relation to the crystalized ligand. Concerning ROC, its indole ring was anchored above the plane of the crystallized ligand with inverted orientation for its substituted methylbutylidene group compared with that of ISO. Similarly, as in the previous two target bindings, the differential ROC/ISO pocket binding modes were proposed for the inverted geometrical configurations of their respective methylbutylidene moieties.

Stability of the MEL/target complex was mediated through combined binding interactions with crucial pocket residues being translated into high docking binding energy (−9.96 Kcal/mol; rRMSD 1.34 Å). Anchoring at the c-Met hinge region was driven through several polar bonds. Double bonding with the Met mainchain NH (2.8 Å/136.3°) and carbonyl group (1.3 Å/143.6°) were mediated via the ligand’s imidazole hydrogen bond donor (NH) as well as oxo-substitution on ligand’s piperazine ring, respectively (Figure 4C). The free hydroxyl group of the ligand’s scaffold illustrated strong hydrogen bond pairing with Pro1158 mainchain carbonyl (2.1 Å/163.7°). The ligand’s imidazolidinone ring was depicted important for further anchoring of MEL at the hinge region through hydrogen bond interaction with the mainchain of its constituting residue, Gly1163 (2.9 Å/109.3°). The terminal hydrophobic part of MEL showed close proximity and favored orientation being inserted in the space between Tyr1230 of the A-loop and the opposite Met1211 residue mediating relevant π-π stacking (3.0–3.8 Å) for the opposite aromatic rings. Hydrophobic interaction with Tyr1230 and Met1211 would contribute to target selectivity as both residues are conserved among only three (c-Met, Mer, Axl) kinases out of the 491 family members [103]. Other hydrophobic non-bonding interactions were illustrated for the MRL’s methylbutylidene moiety towards the non-polar pocket lining residues such as Val1092 and Ala1226.

Binding mode of ROC showed almost comparable residue-wise binding patterns and docking binding energy (−9.78 Kcal/mol; rRMSD 1.25 Å) in relation to those for MEL. Polar hydrogen bonding between the ligand’s imidazole head and Met1160 mainchain (2.1 Å/127.5°) was illustrated for ROC stability at the enzyme hinge region (Figure 4D). The presence of piprazinedione ring in ROC managed to achieve polar contact with the Pro1158 mainchain NH (2.8 Å/141.6°), yet no relevant binding with Met1160 unlike the MEL’s imidazoline ring. The latter differential binding mode highlights the more preferential role of imidazoline for ligand anchoring at c-Met hinge region compared with that of piprazinedione one. Despite differential ROC/MEL-associated orientation of their terminal aromatic rings, ROC exhibited the π-π ring stacking with Tyr1230 sidechain owing to its proximity (3.3 Å) and almost favored orientation. However, ROC lacked relevant π-mediated interaction with Met1211 due to its far side orientation. Interestingly, the additional closeness of the ligand’s methylbutylidene moiety towards Tyr1230 within a 5 Å distance (2.9 Å) permitted relevant CH-π hydrophobic interaction and so, further ligand stability at the A-loop side. This additional non-polar binding would compensate the lower ROC-associated polar interactions with the hinge region residues. Finally, another π–H (3.1 Å) contact between the Gly1163 mainchain NH and the ligand’s imidazole ring further stabilized the ROC at the hinge region of the c-Met pocket.

Moving towards ISO docked pose (Figure 4E), the imidazole ring depicted hydrogen bond with the Met1160 mainchain carbonyl group (2.2 Å/112.3°). On the other hand, the central piperazindione ring was quite retracted from the hinge region achieving a long hydrogen bond with the Met1160 mainchain nitrogen (3.3 Å/127.2°). Unlike its close analogues, ISO lacked relevant polar interaction with hinge Pro1158 owing to the retracted piperazindione scaffold. Additionally, the improper orientation of the ISO’s terminal aromatic ring unfavored π-driven interactions with Tyr1230 of the A-loop side and so as with Met1211. Only single π–H (4.1 Å) contact was depicted between the Gly1163 mainchain NH and the ligand’s imidazole ring that further stabilized ISO at the hinge region of the c-Met pocket. Other hydrophobic non-bonding interactions were also seen for ISO’s methylbutylidene moiety with Val1092 and Ala1226. All of which was correlated to lower docking binding score (−8.35 Kcal/mol; rRMSD 1.11 Å). It is worth noting that the docking binding energy profiles for the three simulated ligands at c-Met pocket were higher than those at PTP-1B and Cdc-25A ones conferring a preferential c-Met-associated affinity for these indole-based alkaloids.

The presented ligand-c-Met docking analysis was validated through furnishing great overlaid conformation for the redocked co-crystallized ligand with low RMSD value (1.56 Å; Figure 4F). Moreover, presenting the simulated indole-based alkaloid with comparable residue-wise binding interactions for several reported c-Met inhibitors further confirmed the presented docking study. Consistency for ligand-directed binding towards Pro1158, Met1160, and/or Tyr1230 was reported with different c-Met inhibitors bearing quinolinylmethyl purine, dihydroquinoline, thiadiazolo [2,3-c]-triazin, and indazole scaffolds showing low-to-sub micromolar c-Met inhibition activities [104,105,106,107,108]. Studies employing quantitative structural activity relationship-aided molecular dynamics design of potent and exquisitely selective c-Met inhibitors bearing heterocyclic fused ring scaffolds [109,110]. These studies highlighted hydrophobic interactions with Val1092, Ala1108, Leu1157, Tyr1159, Met1211, and Tyr1230, as well as hydrogen bonding with Met1160 and Asp1222, particularly for ligands with hydrogen bond donors, for favor stabilization and free binding energy contributions. Moreover, Damghani et al. highlighted critical c-Met inhibitor interactions out of 17 different c-Met complexes running through 30 ns molecular dynamics ensembles [111]. Polar interaction with Pro1158 and Met1160, π-stacking with Tyr1159 and/or Tyr1230, as well as van der Waal contacts with non-polar lining residues all were found critical.

Comparing our MEL-oriented analysis with that reported by Mady et al., showed that our docking findings were more consistent with the c-Met crucial residue-wise binding [31]. The authors reported inverted orientation for MEL inside the c-Met pocket with its imidazole ring being directed towards the A-loop, furnishing π-π stacking with Tyr1230, while extending its ring towards the pocket entrance allowing hydrogen bonding of its imidazolidinone ring with P-loop vicinal residue Ile1084. This inverted pose could not furnish any significant polar interaction with the hinge region residues being crucial for all kinds of c-Met inhibitors including type-I, -1.5, and -II [36,37]. In these regards, their MEL computational results were inappropriate for translating the reported c-Met biological findings. Moreover, the authors applied docking only for MEL making the study lack comprehensive molecular modelling insights regarding the detailed structural-activity relationship with its close analogues which would be beneficial for future lead development and optimization. Here, we further investigate the thermodynamic stability of the three indole-based alkaloids through 100 ns molecular dynamics simulations. This would provide greater insights regarding free binding energies and residue-wise stability/energy contributions under near-physiological conditions as being far more superior over the static or even most sophisticated flexible docking techniques [112].

### 3.4. Molecular Dynamics Simulations

#### 3.4.1. Analysis on Cdc-25A

Molecular dynamics simulations were conducted on the obtained ligand/target poses to explore the thermodynamic nature of indole-based metabolites as well as their relative stability at their biological targets. Monitoring the RMSD trajectories of both the simulated ligand and Cdc-25A protein, across the simulation time, would provide great understanding for the conformational changes and relative stabilities of the ligand–target complexes [47,113,114]. On general bases, RMSD trajectories provide accurate measurement regarding a molecular deviation from its reference structure at the beginning of the molecular dynamics simulations [115]. High protein RMSDs usually correlate to significant conformation alterations and instability, while as for ligand they confer compromised ligand–target affinity and ligand pocket accommodation [116]. Monitored protein C^α^-atom RMSD trajectories depicted typical dynamic behavior and significant stability across the simulated times (Figure 5A). Over the initial time frames, the Cdc-25As RMSDs increased as the constrains were removed at the start of the molecular dynamic simulations, then trajectories attained equilibrated plateau for more than half the simulation times. The illustrated RMSD-directed behavior ensured successful Cdc-25A protein convergence across the 100 ns without further extension as well as the suitability of the minimization and equilibration stages to conduct valid molecular dynamic simulation free from relevant artefacts [117]. Interestingly, the simulated proteins in complex to MEL and related analogues showed lower RMSD fluctuations compared with the apo/unliganded state of Cdc-25A protein. The latter confer preferential stability impact of the bounded ligands on the Cdc-25A protein thermodynamic stability. Concerning comparative ligand–protein stabilities, the steadiest RMSD trajectories were assigned for the Cdc-25A in complex with MEL (average 2.18 ± 0.16 Å) compared with those of ROC (2.24 ± 0.17 Å) and ISO (2.45 ± 0.18 Å), reflecting the preferential MEL-Cdc-25A complex stability.

Regarding the ligand-oriented RMSD tones, limited fluctuations were depicted for MEL (5.03 ± 0.63 Å) and ROS (5.38 ± 0.96 Å) in relation to their reference structure (Figure 5B). Higher tones (6.62 ± 1.12 Å) were assigned for ISO, particularly around 40–70 ns, which highlights significant conformational shift. However, this isomeric ligand managed to achieve its respective dynamic equilibration and RMSD plateau until the end of the simulation run conferring sufficient stability. The latter differential ligand-based behavior was also highlighted through conformational analysis of the simulated ligand at the initial and final time frames. Overlaid frames at 0 ns and 100 ns showed more conformational changes for ISO compared with those for MEL and ROC (Figure 5C). The simulated ISO depicted inversion of its indole-based scaffold towards the catalytic P-loop at the end of the simulation. Additionally, the flexible *C*-terminal loop showed conformational drift at the end of the simulation towards the ISO structure causing the ligand to be tucked within the binding site. The latter conformational drift was less observed with MEL and ROC. It is worth mentioning that the ligand and their respective Cdc-25A RMSD tones were around 1.5-fold differences which further ensured significant convergence of the simulated proteins [118,119].

Further stability analysis was performed by monitoring the Cdc-25A proteins’ Cα-atom RMS-Fluctuation (RMSF) tones across the entire simulation runs to dissect the acquired protein’s flexibility/stability profiles down to their respective amino acid levels. Similar to RMSD, the RMSF trajectories estimated the amino acid-directed dynamic behavior (movement and flexibility) based on their deviations from reference positions [120]. For normalizing the RMSF data across the three simulated models, the difference RMSF (ΔRMSF) trajectories were estimated for each ligand-bound Cdc-25A protein in relation to the apo/unliganded protein form (ΔRMSF = RMSF^apo^ − RMSF^holo^) [121]. Significant structural movement and flexibility were assigned for residues with negative ΔRMSF values, while as relevant residue-wise stability and immobility were corresponding to values equal or above 0.3 cut-offs [81,113,121]. Interestingly, several residue ranges at the core protein structure depicted significant immobility and stability profiles, including Thr348-Ser360 vicinal to the *N*-terminus; Asp383-Gly393 of CH2A motif; His429-Pro438 of the P-loop/active site, Leu465-Glu471 of CH2B motif, and Met488-Glu495 of the carboxy end (Figure 6). The highest immobility profiles were assigned to the P-loop and carboxy end amino acids which highlights the preferential influence of the ligand’s binding on Cdc-25A stability particularly at these pocket lining residues. Inherited flexibility of the *C*-terminal chain is reported with the crystalline structure of Cdc-25A and close homologues phosphatases as well as being consistent with the deposited B-factor values [19,122,123,124].

Depicting high positive ΔRMSFs as represented in Table 2, residues such as Arg385, Tyr386, Pro387, Tyr388, Glu389, catalytic Cys430, Phe432, Ser433, Ser434, Glu435, Arg436, Gly467, Gly468, Tyr469, and Lys470 were considered significant for the ligand binding and stability within the Cdc-25A binding site. Findings were in good agreement with the above-described docking results for the specific residue-wise stability of ligand–Cdc-25A complexes. Notably, higher stability trends were assigned for MEL compared with ROC and ISO, where the residues of the MEL-bound Cdc-25A protein depicted less negative or even higher positive ΔRMSF values for the same structural region (Figure 6 and Table 2). Such dynamic behavior implies a ligand-based preferential Cdc-25A stability, particularly for MEL over those of ROC and ISO, which is also consistent with the obtained RMSD findings. It worth mentioning that very few residue regions of the holo Cdc-25A proteins depicted negative ΔRMSF values, meaning that the RMSF of the apo state is much higher in almost all structural regions. In other words, depicting limited negative ΔRMSF residues confers the significant positive impact of ligand binding on Cdc-25A stability where such influence is beyond the canonical binding site affecting most target regions including the far ones.

In order to understand the nature of ligand–Cdc-25A interaction and estimate the affinity magnitude as well as individual energy contributions, free binding energy calculations were performed [125]. The trajectory-oriented Molecular Mechanics-Poisson Boltzmann Surface Area (MM-PBSA) approach was applied since it is considered of comparable accuracy to Free-Energy Perturbation approaches, yet with much lower computational expenses [56]. To our delight, the three simulated indole-based alkaloids depicted relevant affinity towards the Cdc-25A target with total free-binding energies being preferentially higher for MEL and modest for ISO (Table 3). The latter depicted calculations came in agreement with the preliminary docking results depicting preferential affinity for MEL towards the Cdc-25A binding site. Dissection of total energies into its contributing terms illustrated high prevalence for the Coulomb’s electrostatic potentials over the Lennard-Jones hydrophobic interactions for stabilizing MEL and ROC at their respective complexes. On the other hand, the ISO-Cdc-25A complex showed higher van der Waal energy contributions than those of the electrostatic ones. The latter can be the reason for the conformational shift depicted by ISO’s structural moieties that favored hydrophobic contacts while as losing the polar ones. Interestingly, a direct correlation between the electrostatic potentials and polar solvation energies was illustrated where the earlier considered favored for blinding and the latter not since binding is a solvent substitution process. Both MEL and ROC complexes depicted high electrostatic potentials and polar solvation energies while the opposite was assigned for that of the ISO model. This can be explained by the differential ligand orientation throughout the simulations as well as the shallow solvent exposed nature of the Cdc-25A pocket allowing highly ordered water molecules to exist at the pocket binding surface. In this regard, increasing the hydrophobic nature of the binding ligands as well as incorporating ionizable yet hydrophobic moieties (e.g., tetrazole ring) within their structure can serve in maximizing ligand binding, minimizing unfavored polar solvation energies, and satisfy the polar nature of the Cdc-25A pocket.

Exploring the residue-wise energy contribution within the different ligand–Cdc-25A binding was considered beneficial to pinpoint the key pocket residue important for ligand–protein stability. Findings within Figure 7 showed significant favored energy contributions for the P-loop residues Cys430, Glu431, Phe432, Ser434, Glu435, and Arg436 (−1.26 to −14.33 kJ/mol) within ligand–Cdc-25A binding with higher negative energy values for MEL model. Lower energy terms were assigned for the residues of CH2A compared with those of CH2B motif where Glu471, Phe473, Met474, and Tyr485 depicted significant energies from −1.33 to −3.61 kJ/mol. On the other hand, ISO-Cdc-25A stability is more favored through CH2A residue energy contributions than those of CH2B motif. Notably, both Phe473 and Met474 were preferentially contributing to ROC model stability rather than either MEL or ISO. Several *C*-terminal residues showed high contribution within the ligand–Cdc-25A binding including Asp492, Phe493, and Glu495 with more preferentiality for ISO complex rather than MEL and ROC. The high *C*-terminal energy contributions were consistent with the depicted their high immobility profiles at the ΔRMSF analysis as well as the ISO-tucked *C*-terminal conformation drift. This can further highlight the important role of the *C*-terminal chain for stabilizing ligand binding owing to its close proximity from the Cdc-25A canonical pocket. On the other hand, several *N*-terminal residues (Ala350, Lys352, Lys357) as well as motif’s vicinal amino acids (Ala373, Arg439, and Arg486) exhibited unfavored positive energy contributions conferring repulsive forces against the ligand binding. Owing to polar nature of the latter repulsive residues, structure optimization towards increased hydrophobicity and minimized solvation energies were highlighted beneficial for developing anti-Cdc-25A agents with better binding profiles.

#### 3.4.2. Analysis on PTP-1B

Stability of the simulated ligand-PTP-1B complexes was evaluated through monitoring the RMSD trajectories of both ligand and bound proteins. Steady RMSD trajectories with limited fluctuations were illustrated for the PTP-1B proteins bound with the simulated indole-based alkaloids as well as the co-crystallized ligand, INC (Figure 8A). All bound protein RMSDs were depicted lower than those of the Apo PTP-1B state were the latter depicted higher values (3.08 ± 0.48 Å). The latter comparative holo versus apo RMSD trajectory confirms the role of ligand-binding within the PTP-1B protein stability and compactness. Regarding ligand-oriented comparative RMSD trajectories, both MEL and INC-bound proteins showed the steadiest comparable RMSD tones (1.89 ± 0.15 Å and 1.87 ± 0.17 Å, respectively) regarding the other simulated ligands (2.13 ± 0.20 Å for ROC and 2.16 ± 0.27 Å for ISO-bound proteins). Higher fluctuations were reported with ISO and ROC, particularly around the 50–90 ns time window, conferring relevant conformational changes. Interestingly, all simulated target proteins converge around an average value (~2.18 Å) at the end of the molecular dynamics simulation which ensured relevant protein stability, convergence, and molecular dynamic validity without any need for further time extensions.

Ligand’s stability/conferment within the bound pocket was illustrated through the estimated sole ligand RMSD trajectories. Both MEL and ROC were of the steadiest RMSDs (4.08 ± 0.44 Å and 4.51 ± 0.66 Å, respectively) across most of the molecular dynamics runs, despite of limited fluctuations for ROC at the simulation start (Figure 8B). This would confer significant ligand pocket confinement with limited conformational/orientation changes for these two alkaloids across the whole simulation time frame. Contrarily, ISO depicted higher RMSD tones and significant fluctuations across the 35–55 ns (up to 12 Å) and the last 20 ns (8.57 ± 1.24 Å) of the simulation runs. In turn, this dynamic behavior highlights significant conformational and/or orientational drift for ISO within the PTP-1B binding site. Regarding the co-crystallized ligand, INC RMSD trajectories had significant initial fluctuations around 10 ns and 30 ns up to 10.5 Å; however, the ligand soon attained its respective equilibration plateau around the lowest average RMSDs (3.46 ± 1.03 Å) until the end of the simulation run. Notably, INC attained comparable final RMSD to both MEL and ROC (~3.63 Å) at the end of the molecular dynamic timeframe. It is worth mentioning that the RMSD tones of the thermodynamic stable ligands (MEL, ROC, and INC) were no more than 1.5-fold higher than those of their and their respective PTP-1B proteins which further ensured significant protein convergence and complex stability [118,119].

The above ligand-based RMSD trajectories were confirmed through overlaid frames for the start and end simulations (0 ns and 100 ns) (Figure 8C). Conformational analysis of MEL at those time frames showed limited conformational changes with just a small retraction of its imidazole ring at the catalytic P-loop site, yet deeper insertion of its terminal aromatic scaffold towards the Tyr-P cavity. On the other hand, ROC exhibited relevant stability of its imidazole ring at the catalytic P-loop; however, its indole-based scaffold showed an orientation shift slightly in the opposite direction of Tyr-P site. Such conformational/orientation changes can be the reason for the depicted limited ligand RMSD fluctuations at the beginning of the ROC simulation run. Concerning the isomeric analogue, ISO, the ligand depicted total inverted conformation at the PTP-1B binding site. The imidazole ring of simulated ISO was directed towards the solvent side, while its indole-based hydrophobic scaffold was inserted into the P-loop site. Owing to the high charge density of the catalytic P-loop site, directing the ISO’s indole-based scaffold can compromise to some extent the ligand binding and this can partially reason the high ligand RMSD fluctuations at the end of the simulation run.

Dissecting the protein fluctuation patterns down into its constituting residues, as in term of ΔRMSFs, allowed us to pinpoint the key residues being important for ligand binding as well as grasp the residue-wise behavior/motion of the protein binding pocket and its vicinal loops [126,127]. Except for carboxy terminals and limited residue ranges, almost all PTP-1B-bound secondary structures depicted significant immobility and stability profiles with respective ΔRSMF above the 0.30 cut-offs (Figure 9). The latter implies significant impact of ligand-binding on holo protein stability regarding its apo form which was confirmed in the above-described RMSD tones. Notably, the residue ranges around Asp245-Lys255 and Ser270-Trp291 depicted the greatest inflexibility profiles (ΔRMSF tones up to 1.76 Å) owing to their reported high intermolecular binding and secondary structure compactness [128,129,130]. Nevertheless, the far carboxy terminal residues depicted the highest mobility/fluctuation profiles (ΔRMSF tones down to −2.00 Å) being typical to PTP-1B thermodynamic stability profiles and reported B-factor analysis [90]. On the other hand, the significant immobility profile for the *N*-terminal and vicinal residues highlights the ligand-driven impact on the stability of residues being far from the canonical catalytic site.

Comparative ligand-oriented ΔRMSF analysis assigned MEL-bound protein with the highest positive patterns across almost all its respective residue ranges. Contrarily, the protein’s ΔRMSF in complex with ISO exhibited the lowest trends highlighting a compromised ligand–protein binding pattern across the molecular simulation timeframe. Notably, ISO-bound protein solely depicted a high flexible residue range (Ser28-Arg33), vicinal to Tyr-P pocket residues, which was absent with other bounded proteins dynamics. An explanation was proposed due to the ISO’s dramatic conformational/orientation shift and repulsive intermolecular forces implied towards the newly assigned ligand’s pose at the end of the simulation run, all further emphasized on the district MEL-PTP-1B complex stability compared with that of the ISO model. Focusing on the dynamic behavior of catalytic P-loop and vicinal key structural loops, higher stability/inflexibility profiles were assigned for the Q- and PTR-loop constituting residues compared with other loops. Concerning the selective Tyr-P cleft, significant stability profiles were also assigned for its constituting residues being higher inbound with MEL and INC than other ligands. The latter highlights the relevant anchoring/stability of MEL and INC at this selective sub-pocket and so in turn potential inhibition activity/affinity towards PTP-1B [27,29,89]. Pinpointing the PTP-1B residues with significantly high stability profiles revealed Tyr46, and Arg47 (PTR-loop); Glu115, Ser118, and Lys120 (R-loop); Trp179, Asp181, and Phe182 (WPD-loop); Cys215, Ser216, and Arg221 (loop); Gln262 and Gln266 (Q-loop); and Arg24, His25, Phe52, Arg254, and Met258 of the Tyr-P sub-pocket (Table 4). All highlighted residues were considered important for the ligand binding and stability and in good agreement with the above-described docking results describing ligand anchoring at the PTP-1B models.

Exploring the ligand-PTP-1B binding affinity was proceeded through the MM-PBSA calculations for binding free energies of the ligand-PTP-1B complexes across the simulated trajectories. The simulated indole-based alkaloids came next to the co-crystallized ligand, INC, where the latter exhibited the highest free binding energy (−60.68 kJ/mol) as represented in Table 5. Notably, MEL and successively ROC-bound complexes came second to INC, while as the isomeric alkaloid analogue, ISO, showed the fairest free binding affinity down to −15.02 kJ/mol. Notably, the four simulated PTP-1B complexes depicted superior contributions for the van der Waal energy terms over the electrostatic potentials reaching up to 2-fold for INC and ISO systems and even 4-fold for MEL and ROC ones. Both INC and ISO illustrated high electrostatic energy contributions compared with other simulated ligands, yet such preferential potentials were associated with high polar solvation energies (140.39 and 154.38 kJ/mol).

High solvation penalty for ISO complex can be the reason the inverted orientation/conformation of ISO can impose electrostatic solvation penalty for directing the ligand’s hydrophobic core towards the high charge density of PTP-1B pocket. Additionally, the ISO inverted conformation compared with both MEL and ROC caused ISO to lose beneficial van der Waal hydrophobic energy contributions all translated into poor total binding free energy. Regarding the co-crystallized ligand, the high aromatic ligand functionalities of INC imposed relevant solvation entropy for displacing highly ordered water molecules at the pocket surface. Nevertheless, the deep anchoring of INC functionality towards the more lipophilic Tyr-P pocket allowed partial compensation of solvation penalties through high van der Waal potentials and in turn high total free binding energy.

For both MEL and ROC the favored orientation of the ligand’s imidazole with its hydrogen bond doner/acceptor groups satisfied the polar nature of the P-loop amino acids as well as impose reduced solvation penalties. On the other hand, the higher MEL-associated van der Waal potentiality can be the reason for the deeper anchoring of the ligand’s terminal scaffold, compared with ROC, towards the Tyr-P cleft at the end of the molecular simulation run. Potential structural optimization for MEL can be proceeded through adding extra extended/branched lipophilic scaffolds able to achieve more extension towards the Tyr-P site. This scaffold should also be decorated with polar substitutions in the way that would fulfill the few polar residues lining the surface pockets.

Residue-wise free binding energy contributions (≥−2.00 kJ/mol) were favored for Met258, Gly259 at Tyr-P site; Arg45, Tyr46 of PTR-loop; Lys120 at R-loop; Asp181, Phe182 of WPD-loop; Ile219, Arg221 of catalytic P-loop; and Gln262 at Q-loop (Figure 10). Notably, residues of R-loop of the least energy contribution for ligand binding since only Lys1166 and Lys120 were significantly high negative for MEL system (−2.49 and −5.88 kJ/mol, respectively). On the other hand, the highest favored energy contributions were assigned for Tyr46 (−3.84 to −7.34 kJ/mol), Phe182 (−3.09 to −6.95 kJ/mol), Ile219 (−3.48 to −6.67 kJ/mol), and Arg221 (−2.48 to −6.88 kJ/mol) across the four simulated ligand-PTP-1B complexes. The hydrophobic nature of most binding-favored residues illustrated the van der Waal superior contribution over the electrostatic potentials. Differential residue energy contributions were Lys120 (−5.88 kJ/mol) for MEL; and Tyr-P Met258 (−3.27 and −4.45 kJ/mol) and Gly259 (−2.78 and −4.55 kJ/mol) for MEL and INC systems, respectively. The latter Tyr-P residue contributions highlights the advent of MEL and INC anchoring at the Tyr-P site for their respective targeted binding affinity. Finally, the Asp181 energy terms was different for ISO compared with other ligands since the earlier depicted unfavored repulsive energies (5.58 kJ/mol) in relation to high negative terms for MEL, ROC, and INC (−7.48, −3.59, and −7.98 kJ/mol, respectively). Moreover, the PTR-loop vicinal residue Asp48 depicted a positive repulsive energy term significantly high for ISO (21.65 kJ/mol) and not that much for INC (8.53 kJ/mol) model. This can further highlight the impact of the ISO’s inverted conformation at PTP-1B mediating unfavored polar solvation penalties with hampered ligand binding, while this was partially compensated at the INC system.

#### 3.4.3. Analysis on c-Met Kinase

Analysis of both ligand and bound protein alpha-carbon RMSDs illustrated system convergence and ligand pocket accommodation at the c-Met ATP-binding site. Typical c-Met protein thermodynamic behavior was shown where the protein RMSD tones rapidly attained equilibrium plateau following few initial frames (Figure 11A). The impact of ligand-binding on c-Met protein stability was ensured since the holo c-Met proteins depicted lower RMSD values (2.95 ± 0.05 Å) as well as limited fluctuations compared with the apo/unliganded protein form (4.19 ± 0.68 Å). Interestingly, the indole-based alkaloids were associated with significantly lower protein RMSD trajectories (MEL 2.78 ± 0.28 Å; ROC 2.70 ± 0.29 Å; ISO 2.78 ± 0.40 Å) than those in complex with the simulated co-crystalized ligand, 6TD (6TD 3.54 ± 0.34 Å). Ligand accommodation at the c-Met ATP-binding site was confirmed through monitoring the sole ligand RMSD trajectories in relation to its reference structure at the simulation start. All ligands, including the co-crystallized 6TD, showed rapid equilibrated RMSD tones around an average plateau just after 5 ns of the molecular dynamics simulations runs (Figure 11B). Notably, the ligand RMSDs within the c-Met models were at lower tones and steadier trajectories (MEL 2.26 ± 0.59 Å; ROC 3.19 ± 0.39 Å; ISO 3.31 ± 0.47 Å; 6TD 4.27 ± 0.76 Å) compared with the same ligands at either the Cdc-25A or PTP-1B systems. The latter dynamic behavior highlighted the more preferential binding affinity for the indole-isolated metabolites towards c-Met kinase in relation to the other cancer-associated biotargets. Additionally, limited ligand conformational/orientational shift would be depicted at c-Met pocket compared with the other binding sites.

Comparative conformational analysis at the start and end of the simulation runs illustrated great ligand confinement at the c-Met binding site with limited conformational/orientational drift at the end of the simulation runs (Figure 11C). At the 100 ns time frame, all simulated alkaloid metabolites depicted a relevant confinement of their respective imidazole rings towards the c-Met hinge region while laying their terminal aromatic scaffolds at the DFG motif of the activation-loop. Concerning the simulated co-crystallized ligand, 6TD illustrated significant conformational shift of its terminal indole ring as the A-loop moved inwards by ~6.00 Å. Quite similarly, 6TD almost maintained its stability at the c-Met hinge region where its *p*-methoxy benzene ring showed tilting in the direction of the gatekeeper residue. It is worth noting that all simulated ligands managed to maintain their type-I c-Met inhibitor fashion owing to their confinement within the ATP-binding site until the end of the simulation runs.

Generally, the A-loop segments within the all simulated holo c-Met models were maintained at the space between the ATP-binding site and alpha carbon-helix (α1⁄C-helix) as one of the enzyme’s main regulatory structures [131]. Such depicted A-loop orientation is characteristic among others as the inactive conformation of the unphosphorylated c-Met kinases where A-loop insertion disrupts the salt bridge ionic-lock between Lys1110 and Glu1127 bringing αC-helix inwards allowing phosphorylative activation of Tyr1234/Tyr1235 A-loop residues [132,133,134]. Notably, ligand’s confinement at the c-Met ATP-binding site as well as relevant hydrophobic interactions with the A-loop residues result in A-loop insertion and inward movement for maintaining the above-described c-Met inactive conformation [131]. Moreover, it has been reported that the adoption of A-loop of such described conformation caused the c-Met kinase to be incompatible with productive peptide substrate/ATP-binding [135]. All herein simulated ligand-bound c-Met models maintained this A-loop canonical autoinhibited/inactive conformation across the adopted simulation runs. However, the MEL-c-Met complex adopted a differential conformation for the c-Met’s characteristic hydrophobic spine residue (Met1131, Leu1142, His1202, and Phe1223) assembly being reported for kinase activation [136]. Only, the c-Met protein in complex MEL showed a distorted hydrophobic spin assembly which would provide another level of c-Met autoinhibitory stabilization and potency for MEL in c-Met catalytic machinery inhibition [131].

Highlighting the per-residue contribution within the c-Met protein mobility and fluctuation patterns was proceeded through ΔRMSF analysis [126,127]. Both the far N- and C-terminal residues exhibited higher fluctuation patterns compared with the core structural residues (Figure 12). This was reported as a typical thermodynamic behavior for c-Met and several kinases through molecular dynamics studies and B-factor analysis [109,110,111,137,138]. Except for a singular residue range (Ala1221-Ala1251), the four simulated c-Met ligand-bound models depicted significant immobility and stability patterns with respective ΔRSMFs being above the 0.30 cut-offs. The latter observation highlighted the impact of ligand binding on stabilizing c-Met models as well as further proposed an induced fitting trends for the protein secondary structure. Notably, the highly fluctuating residue range represents several residues of the A-loop where their respective ΔRSMF tones reached down to high negative values (Table 6). This can be consistent with the depicted movement of the flexible loop within the space between ATP-binding site and 1α/C-helix functioning as a pseudo-substrate protein for maintaining the auto-inhibitory c-Met state [131,135]. It is worth mentioning that the A-loop fluctuation patterns were the least for MEL and highest for both ISO and 6TD-bound proteins. This differential A-loop fluctuation pattern highlighted a significant impact of MEL binding on A-loop adopted conformation which can be correlated with the distorted hydrophobic spine being observed only at MEL-bound protein.

To further explore the MEL-hydrophobic spine correlation, we comparatively investigated the hydrophobic spine ΔRMSF values across the ligand–protein simulated complexes. Both Met1131 and Phe1223 of the hydrophobic spine were of the lowest ΔRMSF values for the MEL model compared with those of other simulated ligands (Table 6). These residues at MEL model were either of low positive ΔRMSF (Met1131) or even being negative (Phe1223) compared with the higher positive values for the same residues at other simulated c-Met models. The latter confirmed the MEL-induced distortion of the hydrophobic spine conformation across the simulation run which highlighted another level for MEL-directed of c-Met autoinhibitory stabilization and catalytic machinery inhibition [131]. Moving towards other c-Met key structural loops, P-loop and 1α/C-helix residues depicted the significant stability and inflexibility profiles (ΔRMSF up to 2.84 Å and 3.77 Å, respectively) for the bound indole-based alkaloids compared with other c-Met regions. The latter highlighted the profound stability of these ligand at the ATP-binding site as well as the ability of bound ligands to maintain the inactive c-Met conformation with 1α/C-helix being outwards. On the other hand, the co-crystallized 6TD depicted the lowest P-loop-associated ΔRMSF values which can be correlated to the depicted ligand conformational/orientational shift particularly for its terminal indole scaffold (Figure 11C). Both the gatekeeper (Leu1157) and hinge region residues showed almost comparable stability and ΔRMSF scores across all simulated systems. This is consistent with reported data for all c-Met inhibitors (type-I/-1.5/-II) where ligand anchoring at the hinge region has been considered indispensable [36,37]. Based on the provided ΔRMSF analysis, favored ligand binding and c-Met affinity was ensured for the simulated ligands being almost comparable for the potent c-Met co-crystallized inhibitor, 6TD.

Free binding energy calculations for the simulated ligands highlighted significant affinity for the alkaloid metabolites towards c-Met binding site being comparable to the potent co-crystallized inhibitor (Table 7). Interestingly, binding affinities were comparable to those obtained at Cdc-25A, yet superior for same ligands at the PTP-1B. This was related to the depicted higher ligand’s RMSD fluctuations for the simulated ligands at the PTP-1B binding site. Dissecting the free-binding energies to the constituting energy term illustrated predominant contributions for the hydrophobic van der Waal potentials over the Coulomb’s electrostatic interactions. This van der Waal preferentiality emphasizes on the significant role of ligand’s hydrophobic contacts with A-loop for keeping the latter inserted between ATP-site and 1α/C-helix maintaining the c-Met inactive state. Predominance of the hydrophobic potentials was also consistent with the energy term patterns obtained for the PTP-1B binding highlighting the importance of achieving non-polar contacts with lining residue for attaining ligand stability at both targets. Nevertheless, the ability of the binding ligands to achieve electrostatic interaction with pocket and vicinal residues would rather satisfy the polar characters of the binding pocket [47]. Higher electrostatic free binding energies were assigned to ISO and 6TD complexes which were also associated with higher unfavored repulsive solvation energies. In these regards, increasing the hydrophobic nature of the indole alkaloids while maintaining relevant electrostatic interactions with pocket’s polar residues maximizes ligand binding.

Per-residue energy contributions highlighted the role of c-Met pocket residues and vicinal amino acids within the binding stability of the simulated ligands. P-loop residues Ile1084 (−0.19 to −6.53 kJ/mol), Gly1085 (−0.46 to −1.33 kJ/mol), and Val1092 (−3.28 to −5.10 kJ/mol) were the highest contributions among other constituting residues of this c-Met structural motif (Figure 13). The latter contributions were more relevant for the three indole-based alkaloids than for the co-crystallized 6TD. Highlighting ligand–hinge region indispensable binding, several constituting residues illustrated high energy contributions including: gatekeeper Leu1157 (−3.37 to −5.05 kJ/mol), Tyr1159 (−3.17 to −5.89 kJ/mol), key c-Met inhibitor binder Met1160 (−2.40 to −8.61 kJ/mol), and Gly1163 (−0.51 to −2.20 kJ/mol). Notably, Met1160 was of higher contributions for the three alkaloids over the co-crystallized 6TD. Repulsive forces and positive energy contributions with the hinge region surface residue Asp1164 (2.35 to 6.21 kJ/mol) were proposed significant for directing the simulated ligand inwards, deep into the ATP-binding site. Negligible energy term contributions were assigned for the 1α/C-helix residues since the simulated ligands adopted the type-I fashion of c-Met inhibitors showing no deep anchoring to the back hydrophobic pocket. Finally, negative-valued attractive energy contributions with A-loop non-polar constituting residues as well as positive-valued repulsive ones with the polar ones were the characteristic patterns for all simulated ligand–protein complexes. Residues such as Met1211 (−4.76 to −9.22 kJ/mol), Ala1221 (−0.30 to −1.22 kJ/mol), Asp1222 (0.56 to 7.38 kJ/mol), Phe1223 (−0.18 to −0.59 kJ/mol), Gly1224 (−0.01 to −0.12 kJ/mol), Leu1225 (−0.11 to −1.13 kJ/mol), Asp1228 (1.01 to 7.91 kJ/mol), Tyr1230 (−0.76 to −6.69 kJ/mol), and Asp1231 (0.72 to 1.90 kJ/mol) were of the highest energy contributions among the A-loop residues. Owing to the higher number of the negative-valued attractive hydrophobic residues for ligand binding, the role of non-polar contacts were highlighted for allowing the A-loop to adopt the autoinhibitory conformation. To our delight, all presented per-residue energy contributions were in good agreement with the previously described residue-wise docking poses at c-Met pocket.

### 3.5. Drug-Likeness and Pharmacokinetic Profiling

To our delight, the indole-based alkaloids predicted favored pharmacokinetic parameters and safety profiles acceptable across the ranges related to 95% of already-known therapeutics deposited in the Qik_Prop^®^ database (Table 8). No reported violation of the Lipinski’s R_O5 was depicted where the values are below the assigned thresholds. Moderate lipophilicity profiles accompanied with balanced aqueous solubility were predicted for the isolated metabolites having both Q_PlogS and Q_PlogP around their respective values −3.39 → −2.15 and 1.76 → 2.36. Lower comparative lipophilic profile was assigned for MEL which impacted the compound’s permeation across variable barrier models being lower than those ISO and ROC. Generally, the three isolated metabolites predicted fair membrane penetration indices in relation to Q_PPMDCK and Q_PPCaco parameters (16.76 → 30.63 nm/s and 39.75 → 69.44 nm/s, respectively) translated into moderate predicted %HOA (61% → 74%) with higher preferentiality for ISO and ROC. The latter absorption-related parameters highlighted the potentiality for future structural development and optimization through incorporation of lipophilic functionalities which would also favor ligand–target binding as described at molecular modelling studies.

Safety profiles of the isolated metabolites predicted minimal CNS influence with low Q_PlogBB values (−1.05 → −0.84). However, the impact on animal model mortality was alarming, showing low oral rat LD_50_ (7.89 → 16.23 mg/Kg). Assessing the compound’s inherited mutagenicity throughout the T.E.S.T^®^ analysis illustrated non-mutagenic profile, particularly for MEL. Safety profiles through detrimental effects on the HERG_K_v_11.1 channels and cardiac QT-prolongation were predicted adequate since the isolated metabolites showed Q_PlogHERG values within the standard and acceptable range. The predicted compound blood existence and accumulation were also assessed through values of the human albumin protein-drug binding (Q_PlogK_HSA_). All metabolites showed values between −0.22 and 0.30 being within the acceptable standard cut-off. Notably, higher Q_PlogK_HSA_ values were assigned for MEL (negative values) which was correlated with its respective higher animal lethality compared with its close analogues. In these regards, introducing ionizable functionality with more lipophilic characters (e.g., triazole ring scaffold) to the MEL structure has been proposed beneficial for obtaining a more balanced pharmacokinetic profile and higher safety characteristics. On the other hand, adopting a site-specific targeted drug delivery system would also be considered beneficial. Based on all above findings, the isolated metabolites predicted significant drug-likeness potentiality as well as promising ADME_TOX profiles serving as potential clinical candidates for further lead development and optimization.

## 4. Conclusions

The presented manuscript introduced molecular insights regarding three major indole-based alkaloid metabolites isolated from *Penicillium chrysogenum* strain S003 as anticancer agents. Sophisticated molecular modelling approaches highlighted the preferentiality of MEL over other isolated molecules which is quite congruent with the obtained in vitro study on different solid cancerous cell lines. The study provided, for the first time, valuable molecular aspects regarding the ligand–target affinity towards three cancer-associated biological targets: Cdc-25A, PTP-1B, and c-Met kinase. Investigations regarding both the free binding energies and drug-likeness/pharmacokinetic characteristics of the simulated metabolites were presented significant for directing future drug optimization as well as lead development towards clinical testing and application. Findings present MEL as a potential multi-target anticancer agent with potentiality for future development and optimization. Introducing ionizable functionality with more lipophilic characters was highlighted to obtain more balanced pharmacokinetic profile and higher safety characteristics as well as improve binding affinities through minimizing the unfavored solvation enthalpy.

## Figures and Tables

**Figure 1 metabolites-13-00162-f001:**
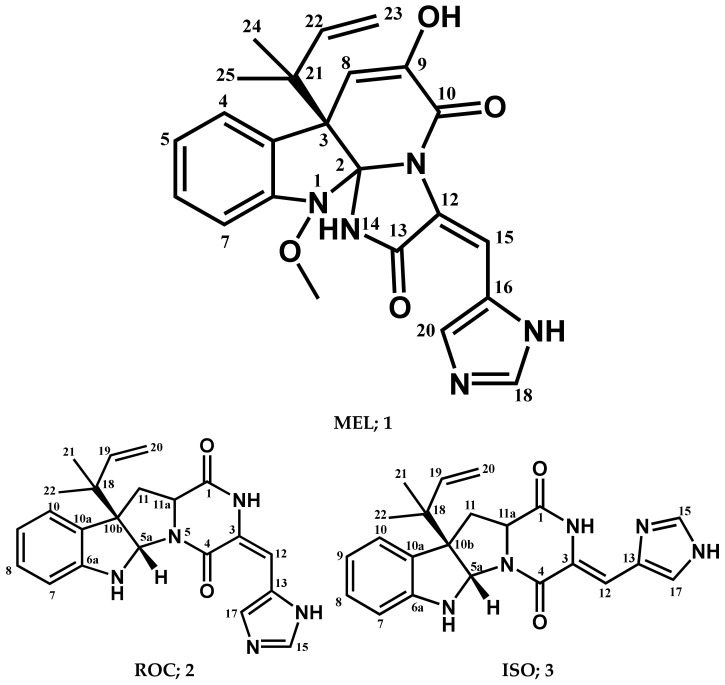
Structures of compounds isolated from *P. chrysogenum* Strain S003.

**Figure 2 metabolites-13-00162-f002:**
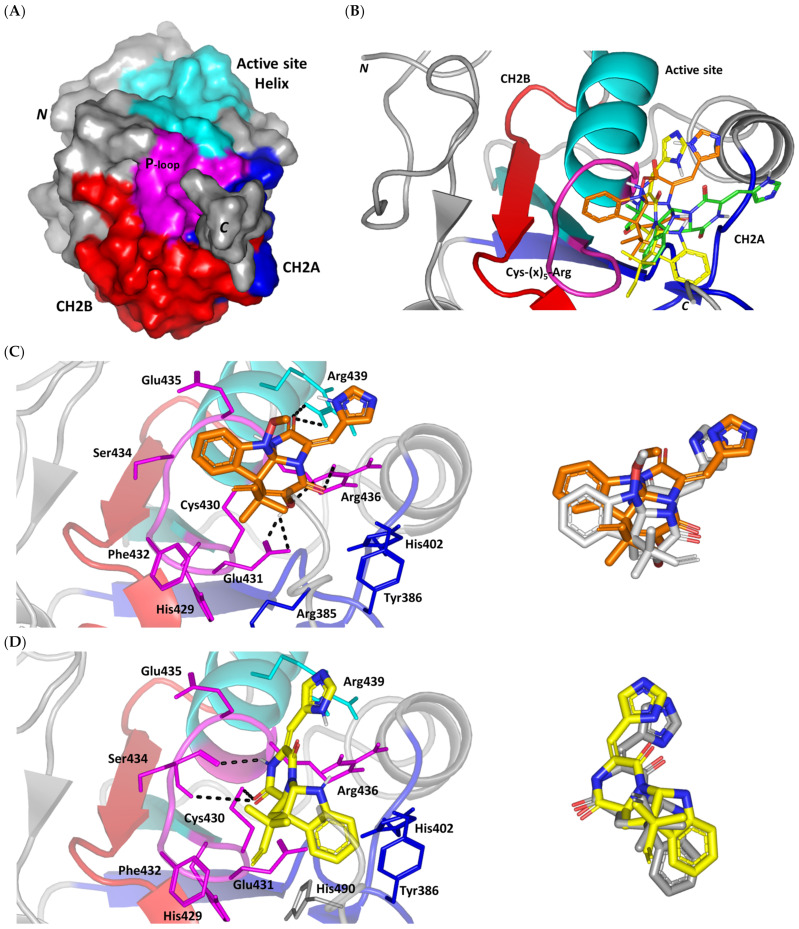
Structure and ligand–target binding interactions at Cdc-25A catalytic site. (**A**) Surface representation of the key structural elements important for Cdc-25A enzyme structure, folding, and catalysis being colored differently; CH2A motif (blue), CH2B motif (red), active site helix (cyan), and catalytic Cys-(x)_5_-Arg motif (magenta). Letters *N* and *C* on the left and towards the viewer denotes the respective amino- and carboxy-terminals of the protein. (**B**) Overlay of docked ligands; MEL (orange lines), ROC (yellow lines), and ISO (green lines) at the Cdc-25A catalytic site (PDB: 1c25) represented as illustrations. (**C**–**E**) Left panels: predicted binding modes of the docked ligands (sticks); (**C**) MEL, (**D**) ROC, (**E**) ISO at Cdc-25A catalytic site. Amino acids within 5 Å radius of bounded ligands are displayed as lines, colored in respect to their position, and labelled with sequence number. Polar interactions (hydrogen bonds) are represented as black dashed lines. Right panels: superimposed docked (colored as described above) and redocked ligands (gray), represented as sticks.

**Figure 3 metabolites-13-00162-f003:**
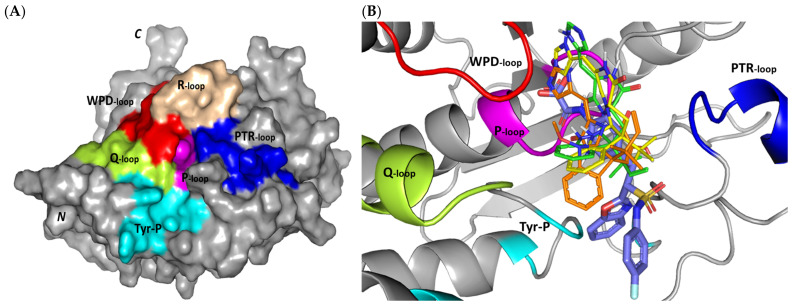
Structure and ligand–target binding interactions at PTP-1B catalytic site. (**A**) Surface representation of the key structural elements important for PTP-1B enzyme structure, folding, and catalysis being colored differently; P-loop catalytic active site (magenta), second aryl phosphate (Tyr-P) binding site (cyan), phosphotyrosine-recognizing loop (blue), R-loop (tint wheat), WPD-loop (red), and Q-loop (limon). Letters *N* and *C* on the left and towards the viewer denotes the respective amino- and carboxy-terminals of the protein. (**B**) Overlay of docked ligands; MEL (orange lines), ROC (yellow lines), ISO (green lines), and co-crystallized IN1834-146C (INC; marine blue sticks) at the PTP-1B catalytic site (PDB: 4i8n) represented as illustrations. (**C**–**E**) Left panels: predicted binding modes of the docked ligands (sticks); (**C**) MEL, (**D**) ROC, (**E**) ISO at PTP-1B binding sites. Amino acids within 5Å radius of bounded ligands are displayed as lines, colored in respect to their position, and labelled with sequence number. Polar interactions (hydrogen bonds) are represented as black dashed lines. Right panels: superimposed docked (colored as described above) and redocked ligands (gray), represented as sticks. (**F**) Superimposing the co-crystallized (marine blue sticks) and redocked (gray sticks) ligands at PTP-1B for validating the adopted docking protocol.

**Figure 4 metabolites-13-00162-f004:**
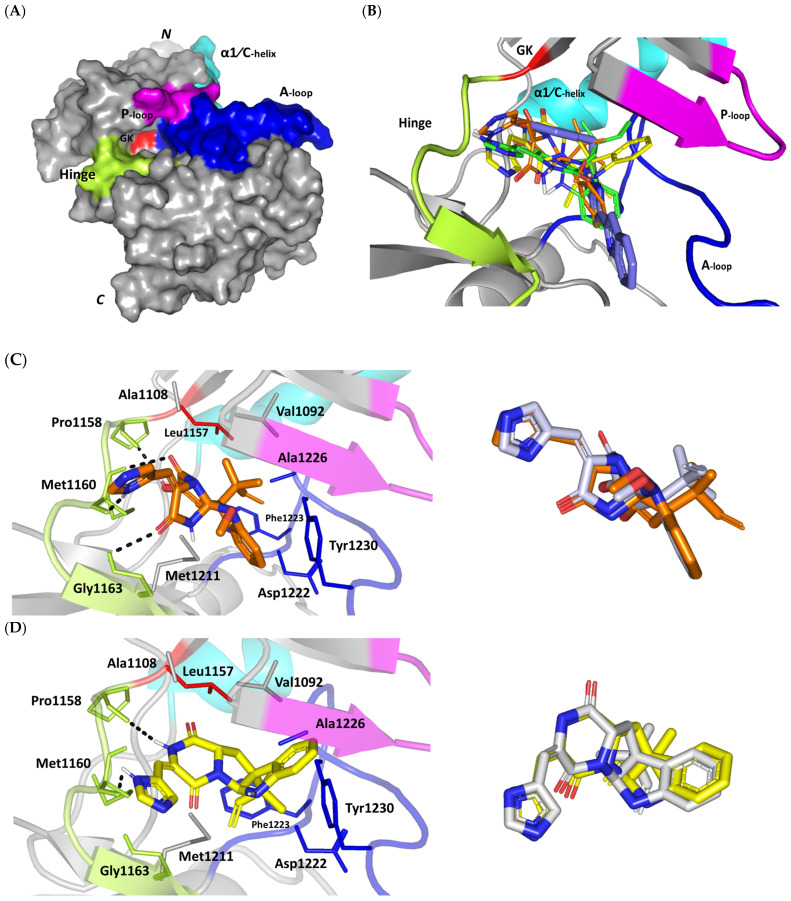
Structure and ligand–target binding interactions at c-Met ATP-binding site. (**A**) Surface representation of the key structural elements important for c-Met enzyme structure, folding, and catalysis being colored differently; hinge region (limon), GK = gatekeeper residue (red), P-loop (magenta), and activation (**A**)-loop (blue). Letters *N* and *C* on the left and towards the viewer denotes the respective amino- and carboxy-terminals of the protein. (**B**) Overlay of docked ligands; MEL (orange lines), ROC (yellow lines), ISO (green lines), and co-crystallized 6TD (marine blue sticks) at the c-Met ATP-binding site (PDB: 5ya5) represented as illustrations. (**C**–**E**) Left panels: predicted binding modes of the docked ligands (sticks): (**C**) MEL, (**D**) ROC, (**E**) ISO at Cdc-25A catalytic site. Amino acids within 5Å radius of bounded ligands are displayed as lines, colored in respect to their position, and labelled with sequence number. Polar interactions (hydrogen bonds) are represented as black dashed lines. Right panels: superimposed docked (colored as described above) and redocked ligands (gray), represented as sticks. (**F**) Superimposing the co-crystallized (marine blue sticks) and redocked (gray sticks) ligands at C-Met kinase for validating the adopted docking protocol.

**Figure 5 metabolites-13-00162-f005:**
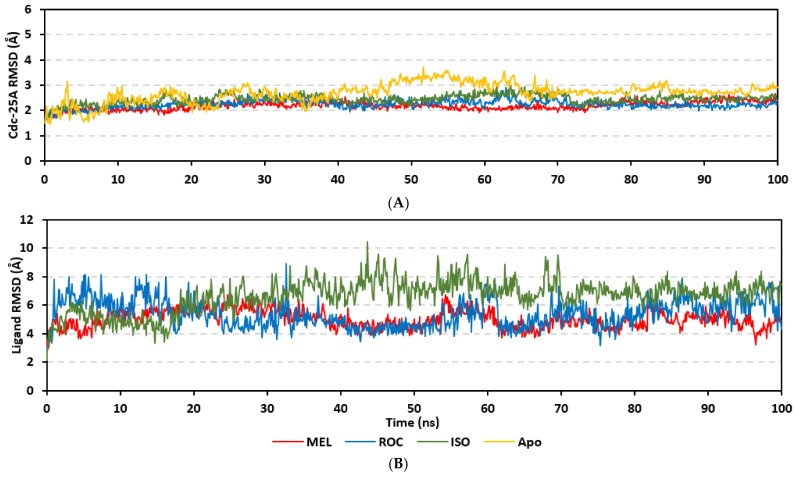
Stability and conformational analysis for the simulated ligands at Cdc-25A complexes. (**A**) Cdc-25A Cα-atom RMSDs; (**B**) ligand backbone RMSDs, in relation to the simulation times (ns). (**C**) Overlaid ligand–Cdc-25A snapshots at 0 ns and 100 ns for MEL, ROC, and ISO at the upper left, upper right, and lower panels, respectively. Ligands (sticks) and respective bound Cdc-15A proteins (illustration) are colored green and red in respect to initial and last extracted frames.

**Figure 6 metabolites-13-00162-f006:**
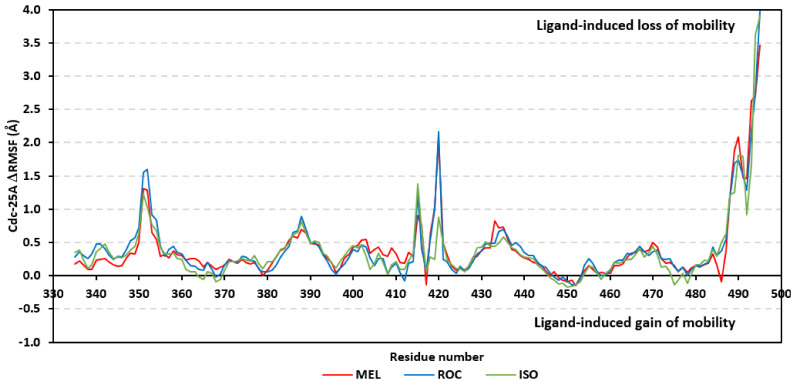
Difference RMSF trajectories analysis for the Cdc-25A-bound proteins across the molecular dynamics simulation. Residue-wise RMSFs of the Cdc-25A protein bounded to MEL, ROC, or ISO in relation to its apo/unliganded state are represented. Key structural motifs at residue ranges; 376–403 (CH2A), 423–450 (active site helix), 429–436 (P-loop), and 459–488 (CH2B).

**Figure 7 metabolites-13-00162-f007:**
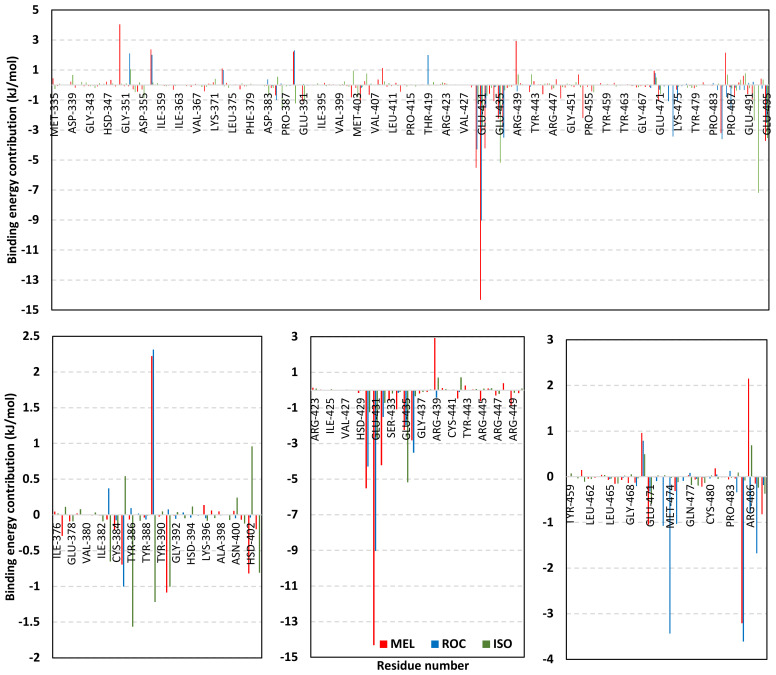
MM-PBSA residue–wise free binding energy contributions for ligand–Cdc-25A complexes. Lower panels are expanded versions of the upper panel representing Cdc-25A key structural motifs; CH2A, active site/P-loop, and CH2B motifs from left to right, respectively.

**Figure 8 metabolites-13-00162-f008:**
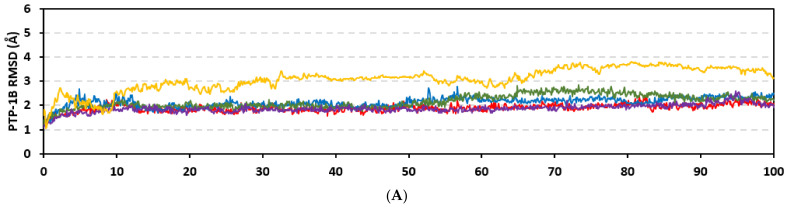
Stability and conformational analysis for the simulated ligands at PTP-1B complexes. (**A**) PTP-1B Cα-atom RMSDs; (**B**) ligand backbone RMSDs, in relation to the simulation times (ns). (**C**) Overlaid ligand-PTP-1B snapshots at 0 ns and 100 ns for MEL, ROC, ISO, and co-crystallized INC at the upper left, upper right, lower left, and lower right panels, respectively. Ligands (sticks) and respective bound PTP-1B proteins (illustration) are colored green and red in respect to initial and last extracted frames.

**Figure 9 metabolites-13-00162-f009:**
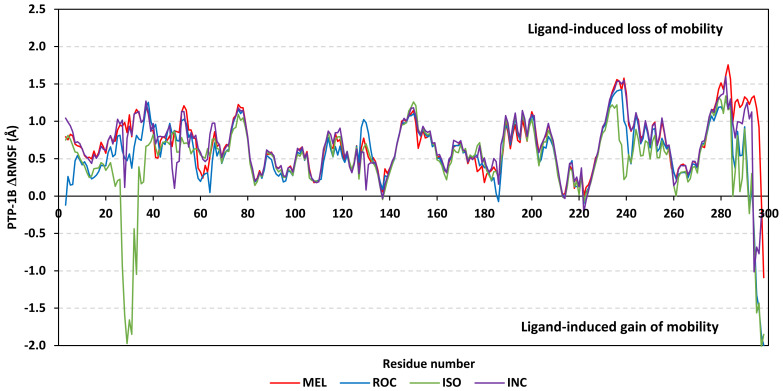
Difference RMSF trajectories analysis for the PTP-1B-bound proteins across the molecular dynamic simulation. Residue-wise RMSFs of the PTP-1B protein bounded to MEL, ROC, ISO, or co-crystalline INC in relation to its apo/unliganded state are represented. Key structural motifs at residue ranges; key structural motifs at residue ranges; 40–46 (PTR-loop), 110–121 (R-loop), 177–185 (WPD-loop), 214–221 (P-loop), and 262–269 (Q-loop).

**Figure 10 metabolites-13-00162-f010:**
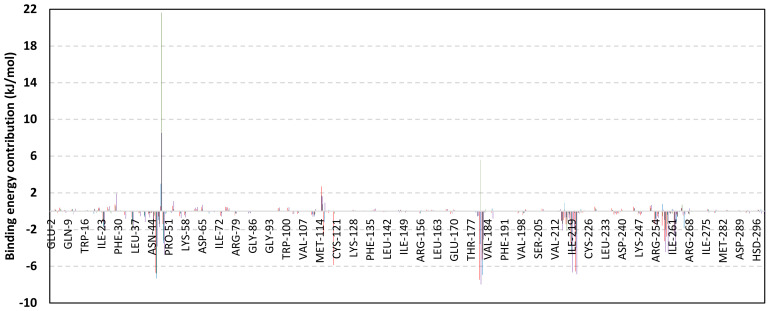
MM-PBSA residue-wise free binding energy contributions for ligand-PTP-1B complexes. Lower panels are expanded versions of the upper panel representing PTP-1B most energy contributing structural loops; Tyr-P/PTR-loop, WPD/P-loop, and Tyr-P/Q-loop from left to right, respectively.

**Figure 11 metabolites-13-00162-f011:**
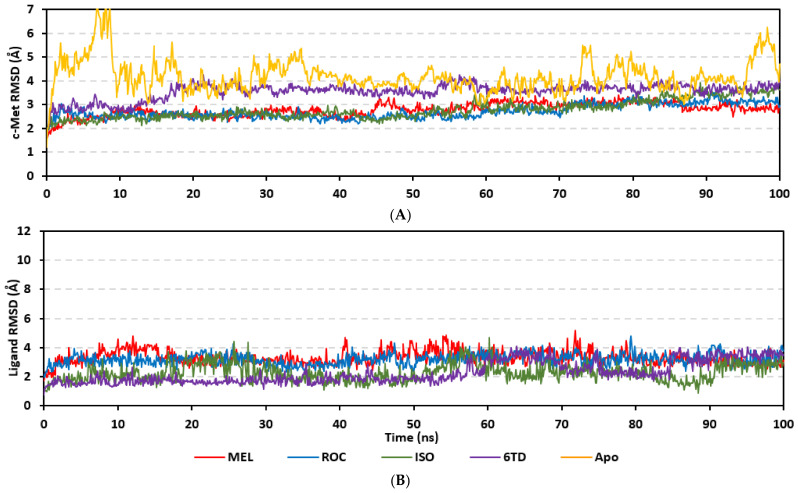
Stability and conformational analysis for the simulated ligands at c-Met complexes. (**A**) c-Met Cα-atom RMSDs; (**B**) ligand backbone RMSDs, in relation to the simulation times (ns). (**C**) Overlaid ligand-c-Met snapshots at 0 ns and 100 ns for MEL, ROC, ISO, and co-crystallized 6TD at the upper left, upper right, lower left, and lower right panels, respectively. Salt bridge ion-lock residues (Lys1110 and Glu1127; lines), hydrophobic spine residues (Met1131, Leu1142, His1202, Phe1223; sticks); ligands (sticks), as well as respective bound c-Met proteins (illustration) are colored green and red in respect to initial and last extracted frames.

**Figure 12 metabolites-13-00162-f012:**
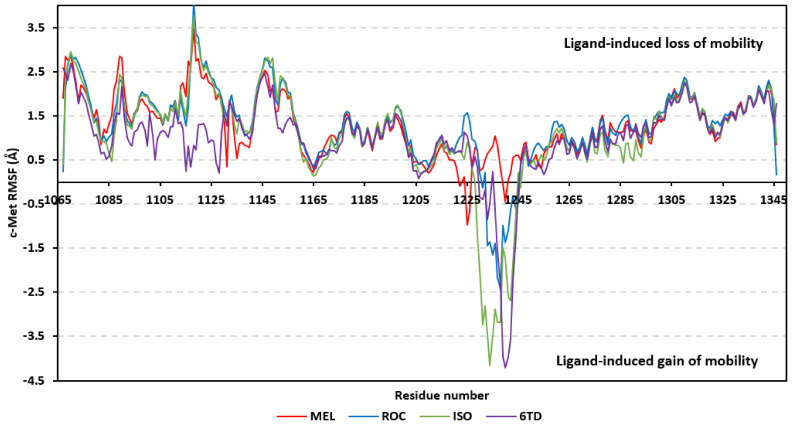
Difference RMSF trajectories analysis for the c-Met-bound proteins across the molecular dynamics simulation. Residue-wise RMSFs of the c-Met protein bounded to MEL, ROC, ISO, or co-crystalline 6TD in relation to its apo/unliganded state are represented. Key structural motifs at residue ranges; key structural motifs at residue ranges; 1085–1090 (P-loop), 1115–1133 (1α/C-helix), 1157 (GK), 1158–1165 (hinge region), and 1221–1251 (A-loop).

**Figure 13 metabolites-13-00162-f013:**
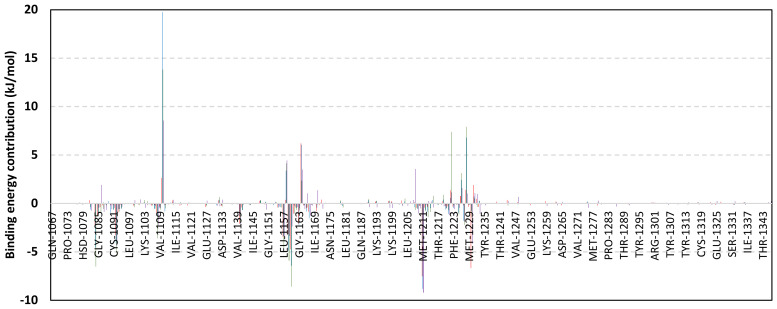
MM-PBSA residue-wise free binding energy contributions for ligand-c-Met complexes. Lower panels are expanded versions of the upper panel representing PTP-1B most energy contributing structural loops: P-loop/1αC-helix, GK/hinge region, and A-loop from left to right, respectively.

**Table 1 metabolites-13-00162-t001:** In vitro activity in µM (±SD) of the isolated compounds against four selected human solid tumor cell lines (*n* = 3).

Cell Type	Cell Line	MEL	ROC	ISO	DOX
Lung cancer	A-549	3.66 ± 0.10	18.70 ± 1.06	20.30 ± 1.06	0.01 ± 0.03
Cervical cancer	HeLa	2.90 ± 0.19	46.97 ± 2.01	53.00 ± 1.36	0.05 ± 0.01
Prostate cancer	DU-145	0.03 ± 0.02	4.80 ± 0.14	17.40 ± 0.44	0.34 ± 0.10
Hepatocellular carcinoma	HepG2	0.10 ± 0.03	7.80 ± 0.69	13.20 ± 0.69	0.92 ± 0.09

**Table 2 metabolites-13-00162-t002:** ΔRMSF of selected residues within the ligand-bound Cdc-25A key structural motifs across the molecular dynamics simulations.

Key Structural Motifs		Residues	MEL	ROC	ISO
**CH2A**		Asp383	** 0.38 **	0.28	** 0.40 **
		Cys384	** 0.40 **	** 0.37 **	** 0.42 **
		Arg385	** 0.53 **	** 0.44 **	** 0.49 **
		Tyr386	** 0.59 **	** 0.65 **	** 0.62 **
		Pro387	** 0.57 **	** 0.68 **	** 0.65 **
		Tyr388	** 0.70 **	** 0.89 **	** 0.81 **
		Glu389	** 0.64 **	** 0.69 **	** 0.64 **
		Tyr390	** 0.49 **	** 0.49 **	** 0.47 **
		Glu391	** 0.47 **	** 0.50 **	** 0.53 **
		Gly392	** 0.47 **	** 0.43 **	** 0.49 **
		Gly393	** 0.33 **	** 0.31 **	** 0.34 **
		Val399	** 0.33 **	0.28	** 0.40 **
		Asn400	** 0.44 **	** 0.40 **	** 0.46 **
		Leu401	** 0.45 **	** 0.36 **	** 0.42 **
		His402	** 0.54 **	** 0.46 **	** 0.47 **
		Met403	** 0.55 **	** 0.43 **	** 0.30 **
**Catalytic site**		Phe428	0.28	0.22	0.27
	**P-loop**	His429	0.29	** 0.33 **	** 0.42 **
		Cys430	** 0.39 **	** 0.38 **	** 0.43 **
		Glu431	** 0.42 **	** 0.48 **	** 0.51 **
		Phe432	** 0.42 **	** 0.49 **	** 0.44 **
		Ser433	** 0.83 **	** 0.49 **	** 0.44 **
		Ser434	** 0.72 **	** 0.66 **	** 0.49 **
		Glu435	** 0.73 **	** 0.70 **	** 0.58 **
		Arg436	** 0.57 **	** 0.60 **	** 0.52 **
		Gly437	** 0.40 **	** 0.45 **	** 0.42 **
		Pro438	** 0.38 **	** 0.50 **	** 0.39 **
		Arg439	** 0.30 **	** 0.45 **	** 0.32 **
**CH2B**		Leu465	** 0.34 **	** 0.32 **	0.24
		Lys466	** 0.35 **	** 0.37 **	** 0.33 **
		Gly467	** 0.40 **	** 0.44 **	** 0.41 **
		Gly468	** 0.36 **	** 0.36 **	0.28
		Tyr469	** 0.38 **	** 0.31 **	** 0.35 **
		Lys470	** 0.50 **	** 0.36 **	** 0.44 **
		Glu471	** 0.43 **	** 0.38 **	** 0.31 **

Residues of significant immobility (ΔRMSF ≥ 0.30 Å cut-off) values are bold and red colored.

**Table 3 metabolites-13-00162-t003:** Free binding and individual energy terms for the simulated ligand–Cdc-25A complexes.

Energy (kJ/mol ± SE)	MEL	ROC	ISO
**van der Waal**	−59.43 ± 26.86	−60.84 ± 38.50	−76.11 ± 29.79
**Electrostatic**	−87.18 ± 39.97	−78.03 ± 28.62	−40.20 ± 15.01
**Solvation; Polar**	70.41 ± 36.218	69.12 ± 41.72	59.05 ± 49.09
**Solvation; SASA**	−8.88 ± 4.77	−7.43 ± 4.82	−10.84 ± 4.45
**Binding energy**	−85.08 ± 17.39	−77.18 ± 11.47	−68.10 ± 13.79

**Table 4 metabolites-13-00162-t004:** ΔRMSF of residues within the ligand-bound PTP-1B key structural loops across the molecular dynamics simulations.

Active Site and Vicinal Structural Loops	Residues	MEL	ROC	ISO	INC
**PTR-loop**	Asn40	** 0.87 **	** 0.90 **	** 0.82 **	** 0.97 **
	Lys41	** 0.52 **	** 0.96 **	** 0.65 **	** 0.70 **
	Asn42	** 0.51 **	** 0.75 **	** 0.54 **	** 0.79 **
	Arg43	** 0.74 **	** 0.52 **	** 0.63 **	** 0.80 **
	Asn44	** 0.79 **	** 0.72 **	** 0.71 **	** 0.79 **
	Arg45	** 0.78 **	** 0.69 **	** 0.73 **	** 0.80 **
	Tyr46	** 0.75 **	** 0.85 **	** 0.72 **	** 0.87 **
	Arg47	** 0.81 **	** 0.97 **	** 0.69 **	** 0.95 **
**R-loop**	Leu110	0.20	0.21	0.19	0.19
	Asn111	** 0.35 **	0.23	** 0.38 **	** 0.38 **
	Arg112	** 0.57 **	** 0.46 **	** 0.61 **	** 0.63 **
	Val113	** 0.72 **	** 0.60 **	** 0.72 **	** 0.73 **
	Met114	** 0.78 **	** 0.78 **	** 0.85 **	** 0.87 **
	Glu115	** 0.66 **	** 0.63 **	** 0.70 **	** 0.82 **
	Lys116	** 0.64 **	** 0.52 **	** 0.53 **	** 0.70 **
	Gly117	** 0.84 **	** 0.66 **	** 0.75 **	** 0.85 **
	Ser118	** 0.73 **	** 0.55 **	** 0.79 **	** 0.85 **
	Leu119	** 0.75 **	** 0.67 **	** 0.79 **	** 0.91 **
	Lys120	** 0.56 **	** 0.52 **	** 0.65 **	** 0.72 **
	Cys121	** 0.48 **	** 0.45 **	** 0.52 **	** 0.51 **
**WPD-loop**	Thr177	** 0.33 **	** 0.54 **	** 0.67 **	** 0.52 **
	Thr178	** 0.36 **	** 0.40 **	** 0.71 **	** 0.67 **
	Trp179	** 0.40 **	** 0.44 **	** 0.53 **	** 0.45 **
	Pro180	0.18	** 0.38 **	** 0.44 **	** 0.51 **
	Asp181	** 0.30 **	** 0.40 **	** 0.34 **	** 0.38 **
	Phe182	** 0.32 **	** 0.30 **	0.28	** 0.33 **
	Gly183	** 0.34 **	0.20	0.21	** 0.37 **
	Val184	** 0.39 **	0.24	** 0.32 **	** 0.50 **
	Pro185	** 0.33 **	0.03	** 0.33 **	** 0.46 **
**Catalytic P-loop**	His214	0.03	−0.01	−0.03	−0.03
	Cys215	0.21	0.17	0.13	0.21
	Ser216	** 0.41 **	** 0.41 **	** 0.37 **	** 0.44 **
	Ala217	** 0.35 **	** 0.47 **	** 0.34 **	** 0.44 **
	Gly218	0.10	0.12	0.13	0.19
	Ile219	0.21	0.13	0.20	0.25
	Gly220	0.14	0.13	0.02	0.13
	Arg221	** 0.34 **	** 0.34 **	0.27	** 0.38 **
	His214	0.03	−0.01	−0.03	−0.03
**Q-loop**	Gln262	** 0.37 **	0.29	0.25	** 0.36 **
	Thr263	** 0.41 **	** 0.31 **	** 0.32 **	** 0.41 **
	Ala264	** 0.43 **	** 0.32 **	** 0.32 **	** 0.40 **
	Asp265	** 0.40 **	** 0.31 **	** 0.33 **	** 0.41 **
	Gln266	0.27	0.24	0.19	0.29
	Leu267	** 0.35 **	** 0.33 **	0.23	** 0.34 **
	Arg268	** 0.47 **	** 0.43 **	** 0.38 **	** 0.47 **
	Phe269	** 0.45 **	** 0.42 **	** 0.39 **	** 0.46 **
**Tyr-P cleft**	Tyr20	** 0.57 **	** 0.40 **	** 0.34 **	** 0.61 **
	Arg24	** 0.74 **	** 0.55 **	0.13	** 0.82 **
	His25	** 0.88 **	** 0.80 **	0.21	** 1.03 **
	Ala27	** 0.93 **	** 0.60 **	−0.93	** 1.02 **
	Phe52	** 1.12 **	** 1.01 **	** 0.78 **	** 1.13 **
	Arg254	** 0.77 **	** 0.64 **	** 0.51 **	** 0.81 **
	Met258	** 0.61 **	** 0.68 **	** 0.65 **	** 0.68 **
	Gly259	** 0.37 **	** 0.50 **	** 0.41 **	** 0.47 **

Residues of significant immobility (ΔRMSF > 0.30 Å cut-off) values are bold and red colored.

**Table 5 metabolites-13-00162-t005:** Free binding and individual energy terms for the simulated ligand-PTP-1B complexes.

Energy (kJ/mol ± SE)	MEL	ROC	ISO	INC
**van der Waal**	−131.05 ± 12.67	−120.20 ± 10.53	−94.19 ± 18.09	−129.63 ± 19.96
**Electrostatic**	−36.77 ± 24.82	−35.34 ± 15.79	−40.20 ± 31.41	−69.59 ± 18.72
**Solvation; Polar**	129.37 ± 39.09	124.63 ± 17.16	140.39 ± 47.46	154.38 ± 18.77
**Solvation; SASA**	−12.74 ± 1.27	−13.81 ± 1.367	−11.02 ± 2.00	−15.84 ± 1.27
**Binding energy**	−51.19 ± 31.56	−44.72 ± 26.39	−15.02 ± 12.78	−60.68 ± 25.99

**Table 6 metabolites-13-00162-t006:** ΔRMSF of residues within the ligand-bound c-Met key structural loops across the molecular dynamics simulations.

Active Site and Vicinal Structural Loops		Residues	MEL	ROC	ISO	INC
**P-loop**		Gly1085	** 1.30 **	** 1.01 **	** 0.67 **	** 0.57 **
		Arg1086	** 1.51 **	** 1.10 **	** 0.47 **	** 0.82 **
		Gly1087	** 2.10 **	** 1.55 **	** 1.04 **	** 1.29 **
		His1088	** 2.29 **	** 1.78 **	** 1.38 **	** 1.57 **
		Phe1089	** 2.84 **	** 2.31 **	** 2.43 **	** 1.53 **
		Gly1090	** 2.82 **	** 2.27 **	** 2.35 **	** 2.17 **
**1α/C-helix**		Ile1115	** 1.94 **	** 1.27 **	** 1.48 **	** 0.23 **
		Thr1116	** 2.74 **	** 1.68 **	** 2.07 **	** 0.81 **
		Asp1117	** 2.60 **	** 2.68 **	** 2.63 **	** 0.33 **
		Ile1118	** 3.47 **	** 4.05 **	** 3.77 **	** 0.83 **
		Gly1119	** 2.74 **	** 3.36 **	** 3.15 **	** 0.72 **
		Glu1120	** 2.78 **	** 3.26 **	** 3.14 **	** 1.30 **
		Val1121	** 2.37 **	** 2.71 **	** 2.69 **	** 1.31 **
		Ser1122	** 2.33 **	** 2.60 **	** 2.53 **	** 1.32 **
		Gln1123	** 2.46 **	** 2.74 **	** 2.63 **	** 1.17 **
		Phe1124	** 2.27 **	** 2.54 **	** 2.49 **	** 0.89 **
		Leu1125	** 2.22 **	** 2.36 **	** 2.34 **	** 0.95 **
		Thr1126	** 2.14 **	** 2.33 **	** 2.18 **	** 0.92 **
		Glu1127	** 1.87 **	** 2.15 **	** 1.96 **	** 0.44 **
		Gly1128	** 1.98 **	** 2.07 **	** 2.00 **	** 0.21 **
		Ile1129	** 1.91 **	** 1.88 **	** 1.72 **	** 1.13 **
		Ile1130	** 1.52 **	** 1.69 **	** 1.40 **	** 1.56 **
		**Met1131**	** 0.34 **	** 1.48 **	** 1.39 **	** 1.28 **
		Lys1132	** 1.85 **	** 1.78 **	** 1.69 **	** 1.83 **
		Asp1133	** 1.71 **	** 1.97 **	** 1.67 **	** 1.73 **
		**Leu1142**	** 1.66 **	** 1.79 **	** 1.73 **	** 1.57 **
**GK**		Leu1157	** 1.43 **	** 1.55 **	** 1.49 **	** 1.28 **
**Hinge region**		Pro1158	** 1.26 **	** 1.39 **	** 1.34 **	** 1.32 **
		Tyr1159	** 1.12 **	** 1.21 **	** 1.12 **	** 1.10 **
		Met1160	** 0.68 **	** 0.86 **	** 0.65 **	** 0.88 **
		Lys1161	** 0.61 **	** 0.85 **	** 0.46 **	** 0.89 **
		His1162	** 0.53 **	** 0.67 **	** 0.54 **	** 0.73 **
		Gly1163	** 0.42 **	** 0.52 **	** 0.33 **	** 0.58 **
		Asp1164	0.28	** 0.43 **	0.23	** 0.43 **
		Leu1165	0.22	** 0.34 **	0.14	0.28
		**His1202**	** 0.62 **	** 0.81 **	** 0.66 **	** 0.55 **
		Ala1221	** 0.33 **	** 0.86 **	** 0.68 **	** 0.67 **
	**DFG motif**	Asp1222	−0.10	** 1.01 **	** 0.70 **	** 0.71 **
		**Phe1223**	−0.01	** 1.07 **	** 0.63 **	** 0.68 **
		Gly1224	0.11	** 1.49 **	** 0.51 **	** 1.13 **
**A-loop**		Leu1225	−0.97	** 1.57 **	** 0.90 **	** 1.04 **
		Ala1226	−0.66	** 1.37 **	** 0.75 **	** 0.77 **
		Arg1227	** 0.36 **	** 0.98 **	** 0.31 **	** 0.30 **
		Asp1228	** 0.77 **	** 0.92 **	−0.01	** 0.57 **
		Met1229	** 0.72 **	** 0.70 **	−1.23	** 0.34 **
		Tyr1230	** 0.26 **	0.20	−1.99	−0.33
		Asp1231	** 0.30 **	−0.12	−3.24	−0.39
		Lys1232	** 0.49 **	0.22	−2.82	−0.19
		Glu1233	** 0.63 **	−1.44	−3.44	−0.85
		Tyr1234	** 0.70 **	−1.36	−4.15	−0.50
		Tyr1235	** 0.81 **	−1.65	−3.48	0.23
		Ser1236	** 1.04 **	−1.40	−2.88	−0.64
		Val1237	** 0.83 **	−2.16	−3.18	−1.58
		His1238	** 0.31 **	−2.42	−3.17	−2.32
		Asn1239	−0.04	−0.99	−1.49	−3.95
		Lys1240	−0.45	−1.37	−1.71	−4.22
		Thr1241	0.06	−1.09	−2.59	−3.97
		Gly1242	0.20	−0.63	−2.69	−3.55
		Ala1243	** 0.56 **	−0.32	−1.84	−2.23
		Lys1244	** 0.61 **	−0.62	−0.68	−1.22
		Leu1245	** 0.60 **	−0.48	0.21	−0.22
		Pro1246	** 0.49 **	** 0.37 **	−0.13	** 0.43 **
		Val1247	** 0.84 **	** 0.69 **	** 0.56 **	** 0.70 **
		Lys1248	** 0.91 **	** 0.69 **	** 0.73 **	** 0.85 **
		Trp1249	** 0.52 **	** 0.37 **	** 0.43 **	** 0.48 **
		Met1250	** 0.47 **	** 0.50 **	** 0.41 **	** 0.35 **
		Ala1251	** 0.50 **	** 0.72 **	** 0.50 **	** 0.30 **

Residues of significant immobility (ΔRMSF > 0.30 Å cut-off) values are bold and red colored. Yellow highlighted bold amino acids denote the hydrophobic spine residues.

**Table 7 metabolites-13-00162-t007:** Free binding and individual energy terms for the simulated ligand-c-Met complexes.

Energy (kJ/mol ± SE)	MEL	ROC	ISO	6TD
**van der Waal**	−146.39 ± 19.06	−171.95 ± 15.78	−173.08 ± 14.38	−175.90 ± 19.17
**Electrostatic**	−16.02 ± 11.72	−15.25 ± 9.46	−31.70 ± 3.82	−30.22 ± 9.00
**Solvation; Polar**	103.31 ± 22.21	133.68 ± 15.86	147.32 ± 12.43	145.35 ± 16.59
**Solvation; SASA**	−17.03 ± 2.32	−17.79 ± 1.18	−17.90 ± 1.10	−17.57 ± 0.82
**Binding energy**	−76.13 ± 10.33	−71.31 ± 22.66	−75.36 ± 17.63	−78.34 ± 11.13

**Table 8 metabolites-13-00162-t008:** Predicted pharmacokinetic and safety profiling (ADME_TOX) ^a^ for the indole-isolated metabolites.

Ligand	“R-O5” HBD HBA θ SASA MW Violation	QPlogP −2.0 → 6.5	QPlogS mol/dm^3^ −6.5 → 0.5	QPPCaco nm/s <25 Poor >500 Great	QPPMDCK nm/s <25 Poor >500 Great	QPlogBB −3.0 → 1.2	QPlogK_HSA_ −1.5 → 1.5	%HOA <25% Poor >80% Great	QPlogHERG >−5.0	Oral Rat LD_50_ μg/Kg	AMES Mutagensis
**MEL**	3 5 4 629.64 433.47 0	1.75	−2.15	39.75	16.76	−1.05	−0.22	61.05	−5.48	7.89	Negative (0.16)
**ROC**	3 3 3 646.57 389.45 0	2.27	−3.39	65.75	28.88	−0.80	0.29	72.77	−5.93	16.23	Negative (0.26)
**ISO**	3 3 3 630.24 389.45 0	2.36	−3.66	69.44	30.63	−0.84	0.30	73.73	−6.33	16.23	Negative (0.26)

*^a^* Values or ranges being recommended-accepted are provided by Qik_Prop^®.^

## Data Availability

Data are available in the manuscript.

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
