# Peer review of "Molecular and Biological Investigation of Isolated Marine Fungal Metabolites as Anticancer Agents: A Multi-Target Approach"

_metabolites, 2023, doi:10.3390/metabo13020162_

Round 1

Reviewer 1 Report

The authors presented the molecular docking simulation of three indole alkaloids (1 to 3) isolated from Penicillium chrysogenum strain S003. However, there are many deficiencies in this manuscript. Since compounds 1 to 3 are known compounds, they have no novelty. The authors also discuss the results of the cytotoxic activity of these known compounds in terms of molecular docking simulations, but since the absolute configuration of compounds 1 to 3 is unknown, the results are meaningless. From the viewpoint of medicinal chemistry, the authors should synthesize derivatives of compounds 1 to 3 and create compounds with stronger cytotoxic activity. And if molecular docking simulations are used to consider the results, I think the novelty of this research will increase. Therefore, this manuscript is not recommended for publication. Listed below are my specific comments.

1.       Line 33; The "Abstract" text is redundant. The authors should describe the results obtained in this study more succinctly.

2.       Line 58; The "Introduction" text is redundant. The authors should briefly describe the background of this study, examples of previously reported studies, the novelty of the results obtained in this study, and their relevance to the authors' previous work.

3.       A listing of the MS and NMR instruments used in this experiment should be added to the experimental section.

4.       Line 199; The authors should provide detailed information on the extraction, isolation and purification of compounds 1 to 3. In particular, the details of column chromatography and HPLC separation conditions and the yield information of compounds 1 to 3 should be added.

5.       Line 207; Compound 1 has five nitrogens. Therefore, from the nitrogen rule, its molecular weight is an odd number (MW: 433). However, the positive-FAB-MS data for compound 1 shows an even molecular weight (MW: 432). The authors should confirm this point.

6.       Compounds 1 to 3 are known compounds. The authors should compare not only the MS and NMR data of these compounds, but also the specific rotation, IR and UV spectral data. In particular, the specific rotation data are important for distinguishing whether compounds 1 to 3 are optically active or racemate. Therefore, the authors should add the specific rotation, IR and UV spectral data of compounds 1 to 3.

7.       This manuscript contains no information on the stereochemistry of compounds 1 to 3. In particular, if the absolute configuration of compounds 1 to 3 has not been determined, the authors should determine it. Also, a molecular docking simulation should be performed based on the results. Without information on the exact absolute configuration of compounds 1 to 3, the results of molecular docking simulations would be meaningless.

8.       Were the authors only able to obtain three compounds in this study? I believe that P. chrysogenum contains many analogous compounds other than these. The authors should comment on the points above.

9.       Line 315; The description about "ODS silica gel column" is missing.

10.    Line 334, Table 1; The number of cancer cells listed in the title does not match that in the table. Also, no information on positive controls is found.

Author Response

Dear Respected Editor-in-Chief of Metabolites

The authors appreciate the constructive criticism and the valuable suggestions of the reviewers and the editorial board concerning our manuscript “Metabolites-2122385”, under title of “Molecular and Biological Investigation of Isolated Marine Fungal Metabolites as Anticancer Agents: A Multi-Target Approach’’

Hereby, the point-by-point reply for the raised corrections and suggestions by the reviewer-1. The authors confirm that they made all the required changes by the referees and were highlighted yellow in the manuscript.

I hope the manuscript in its present form is eligible for publication in your journal.

Please find the attached PDF file.

Reviewer 2 Report

The manuscript submitted by Bogari and coworkers describes firstly the isolation of 3 metabolites from a strain of Penicillium chrysogenum which identity has been confirmed by comparison of spectroscopic data with those reported in literature as said by the authors. Unfortunately, supplementary materials have not been furnished. Thus, I cannot evaluate the purity of the isolated metabolites 1-3 that were tested against cancer cell lines. Please, provide NMR and HR MS spectra.

Moreover, the 3 metabolites were demonstrated to act as anticancer agents and molecular insights on their putative mechanism of action were provided by computational studies. This study provided molecular aspects regarding the ligand-target affinity towards three cancer-associated biological targets of MEL, the most active compound in the series.

The paper is interesting and well organized too; it is rich of experimental details and focus the attention on the multitarget approach to fight cancer. The authors are invited to carry out some modifications such as supporting information and the others listed below; then it could be accepted for publication.

Line 75: I suggest adding an introduction sentence on multitarget approach after dot. “The design of multitarget drugs which are agents that can simultaneously interact with multiple and different biological targets for the treatment of multifactorial pathologies represents a novel interesting approach and a new challenge in medicinal chemistry” (this reference could be cited 10.3390/md19100535)

From line 102 to line 175: this part is too long for an introduction. The role of Penicillium and its secondary metabolites should be summarized without move out the attention of the readers.

Section 2.2: further information about extraction procedure as well as chromatographic separation must be added.

Line 314: crude

Table 1: 4 or 5 cancer cell lines?? Please correct

Line 604: explore

Author Response

Dear Respected Editor-in-Chief of Metabolites

The authors appreciate the constructive criticism and the valuable suggestions of the reviewers and the editorial board concerning our manuscript “Metabolites-2122385”, under title of “Molecular and Biological Investigation of Isolated Marine Fungal Metabolites as Anticancer Agents: A Multi-Target Approach’’

Hereby, the point-by-point reply for the raised corrections and suggestions by the reviewer-2. The authors confirm that they made all the required changes by the referees and were highlighted yellow in the manuscript.

I hope the manuscript in its present form is eligible for publication in your journal.

Please find the attached PDF file.

Round 2

Reviewer 1 Report

This revised manuscript has been modified according to the reviewer’s comments. It is acceptable for publication.

Reviewer 2 Report

The authors carefully revised the manuscript according to comments raised by the referee. It can be accepted for publication.

Adjust reference 7, it lacks number of page (535)